# Full-scale evaluation of methane production under oxic conditions in a mesotrophic lake

D. Donis [1], S. Flury[1,2], A. Stöckli[3], J.E. Spangenberg[4], D. Vachon [1] & D.F. McGinnis [1]

Oxic lake surface waters are frequently oversaturated with methane ($CH_4$). The contribution to the global $CH_4$ cycle is significant, thus leading to an increasing number of studies and stimulating debates. Here we show, using a mass balance, on a temperate, mesotrophic lake, that ~90% of $CH_4$ emissions to the atmosphere is due to $CH_4$ produced within the oxic surface mixed layer (SML) during the stratified period, while the often observed $CH_4$ maximum at the thermocline represents only a physically driven accumulation. Negligible surface $CH_4$ oxidation suggests that the produced $110 \pm 60\ nmol\ CH_4\ L^{-1}\ d^{-1}$ efficiently escapes to the atmosphere. Stable carbon isotope ratios indicate that $CH_4$ in the SML is distinct from sedimentary $CH_4$ production, suggesting alternative pathways and precursors. Our approach reveals $CH_4$ production in the epilimnion that is currently overlooked, and that research on possible mechanisms behind the methane paradox should additionally focus on the lake surface layer.

[1] Aquatic Physics Group, Department F.-A. Forel for Environmental and Aquatic Sciences (DEFSE), Faculty of Science, University of Geneva, Boulevard Carl Vogt 66, 1211 Geneva, Switzerland. [2] Stream Biofilm and Ecosystem Research Laboratory, Institute of Environmental Engineering, School of Architecture, Civil and Environmental Engineering, Ecole Polytechnique Fédérale de Lausanne, Station 2, 1015 Lausanne, Switzerland. [3] Canton Argovia, Department of Civil Engineering, Transportation and Environment, Entfelderstrasse 22, 5001 Aarau, Switzerland. [4] Institute of Earth Surface Dynamics (IDYST), University of Lausanne, Building GEOPOLIS, 1015 Lausanne, Switzerland. Correspondence and requests for materials should be addressed to D.D. (email: daphne.donis@unige.ch) or to D.F.M. (email: daniel.mcginnis@unige.ch)

The appearance of methane ($CH_4$) within oxic surface water of lakes, aka the methane paradox, is an increasingly controversial topic. Normally produced under anoxic conditions, the oversaturation of $CH_4$ in oxic surface waters has been reported for decades in both lakes and oceans[1,2]. While $CH_4$ has been monitored in hypolimnion of lakes for years, it was most often neglected in the surface layer. However, metalimnetic $CH_4$ maxima, thought to be the most intense location for oxic water $CH_4$ production, were found in a number of oligotrophic to mesotrophic lakes including Lake Stechlin, Germany[3], Lake Lugano, Switzerland[4], and Lake Biwa, Japan[5], with concentrations several orders of magnitude higher than $CH_4$ maxima reported for ocean surface oxic waters[6].

While lateral transport from the littoral zone may play an important role for $CH_4$ accumulation in the metalimnion[7,8], mesocosm experiments have convincingly shown that substantial $CH_4$ production can occur in oxic freshwaters[9]. Additionally, a growing number of studies have suggested several pathways leading to $CH_4$ production under aerobic conditions[10,11]. In lakes, a link between the methane paradox origin and algae has been hypothesized given the often observed overlap of the metalimnetic $CH_4$ maxima with oxygen oversaturation and chlorophyll maxima[6]. Other postulations for the presence of $CH_4$ in oxygenated waters include: anoxic micro-niches[12,13], algal metabolites with methionine as a possible precursor[14], and $CH_4$ as a by-product of methylphosphonate (MPn) decomposition[15]. It is plausible that multiple sources act to produce this phenomenon, and that these may vary from lake-to-lake and may be trophic- and/or light-dependent.

The $CH_4$ produced anaerobically in sediments of stratified lakes is efficiently removed by oxidation processes within the lake interior, limiting its evasion to the atmosphere[16,17]. The occurrence of $CH_4$ in oxic surface waters, however, bypasses diffusive limitations to a large extent as it places a $CH_4$ source close to the water surface, intensifying fluxes to the atmosphere[6,18]. Furthermore, the often observed absence or inhibition of $CH_4$ oxidizers in the epilimnion of lakes[19,20] is likely to be particularly significant in this context, indirectly acting to sustain high $CH_4$ concentrations and subsequent emissions. While there is an increasing number of publications on "oxic" methane production (OMP; in the sense of Tang et al.[6], i.e., "without inferring whether the biochemical pathway itself requires oxygen") in lakes, no studies have so far addressed the associated rates under in situ conditions.

In 2015, a distinct $CH_4$ peak was discovered in the oxic thermocline of mesotrophic Lake Hallwil (Switzerland) along with elevated and sustained $CH_4$ concentrations in the surface layer with no clear indications as to their origins. In this study, we quantify the $CH_4$ bulk sources in Lake Hallwil's oxic surface layer using a detailed mass balance approach (Fig. 1) combined with in situ incubation experiments and isotopic evaluations. We come to the unprecedented conclusion that most of the $CH_4$ production actually takes place in the surface mixed layer (SML) (i.e., epilimnion), contrasting the often suggested metalimnetic production. This significant source of $CH_4$ is in direct contact with the atmosphere, implying that lake surface waters may be an important but overlooked $CH_4$ production site.

## Results

**Observations.** We studied Lake Hallwil (Canton Argovia, Switzerland) in 2015–2016 with the goal of isolating the key $CH_4$ sources, sinks, and quantifying production rates as summarized in Fig. 1 (for sampling locations see Supplementary Fig. 1).

**System description.** Mean total phosphorous concentrations in Lake Hallwil are in the range of 10–20 mg m$^{-3}$ [21]. After rigorous restoration measures for the past 30 years, the lake reached a mesotrophic state in 2008 (Supplementary Note 1). The re-oligotrophication process was supported since 1986 by the installation of a hypolimnetic aeration system, placed on the lake bed at ~46 m depth. The air/oxygen flow rate of the system is regulated such that, while preventing anoxic conditions in the deep water, the rising bubble plume does not affect the stratification of the water column in summer[22]. The aggressive restoration measures were extremely effective, resulting in a re-oligotrophication of the lake where bottom waters remain near completely oxic and preventing methane ebullition from developing in the hypolimnetic sediment (Supplementary Fig. 2).

The seasonal evolution of methane in the fully oxic water column is shown on Fig. 2a, b for the years 2015 and 2016, respectively, as well as chlorophyll $a$ (Chl-$a$) (Fig. 2c, d) and temperature (Fig. 2e, f). The $CH_4$ increase is particularly strong within the stratified metalimnion (~5–15 m depth) (0.4 µmol L$^{-1}$ at 6 m in June; 0.5 µmol L$^{-1}$ at 8 m in July; 0.75 µmol L$^{-1}$ at 7 m in August 2016), with the buildup concomitant with the onset of summer stratification (Fig. 2e, f). $CH_4$ concentrations between 25 and 45 m depth at the lake center were low (<0.05 µmol L$^{-1}$). The

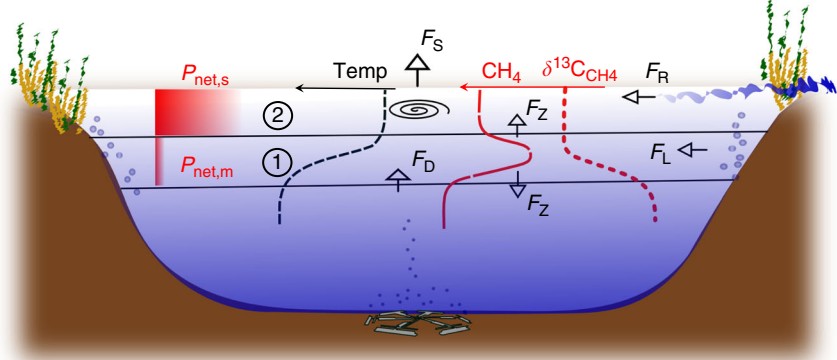

**Fig. 1** Conceptual schematic of the $CH_4$ budget in mesotrophic Lake Hallwil. $CH_4$ mass balance components: evasion to the atmosphere ($F_S$), interior turbulent diffusion ($F_z$), transport from the aeration system ($F_D$), lateral transport ($F_L$), and river input rate ($F_R = Q_R \times C_R$). The case study (Lake Hallwil, Switzerland) was divided into zone 1 (metalimnion) and zone 2 (surface mixed layer). The mass balance reveals that average dissolved $CH_4$ concentration in the summer shows a $CH_4$ metalimnetic maximum concentration (zone 1), however with low production rates ($P_{net,m}$). The highest $CH_4$ production rates ($P_{net,s}$) are actually at the surface (zone 2). The turbulent gas exchange at the lake surface acts to mitigate the zone 2 $CH_4$ concentrations by enhancing outgassing, while the metalimnion gas exchanges are driven by turbulent diffusion

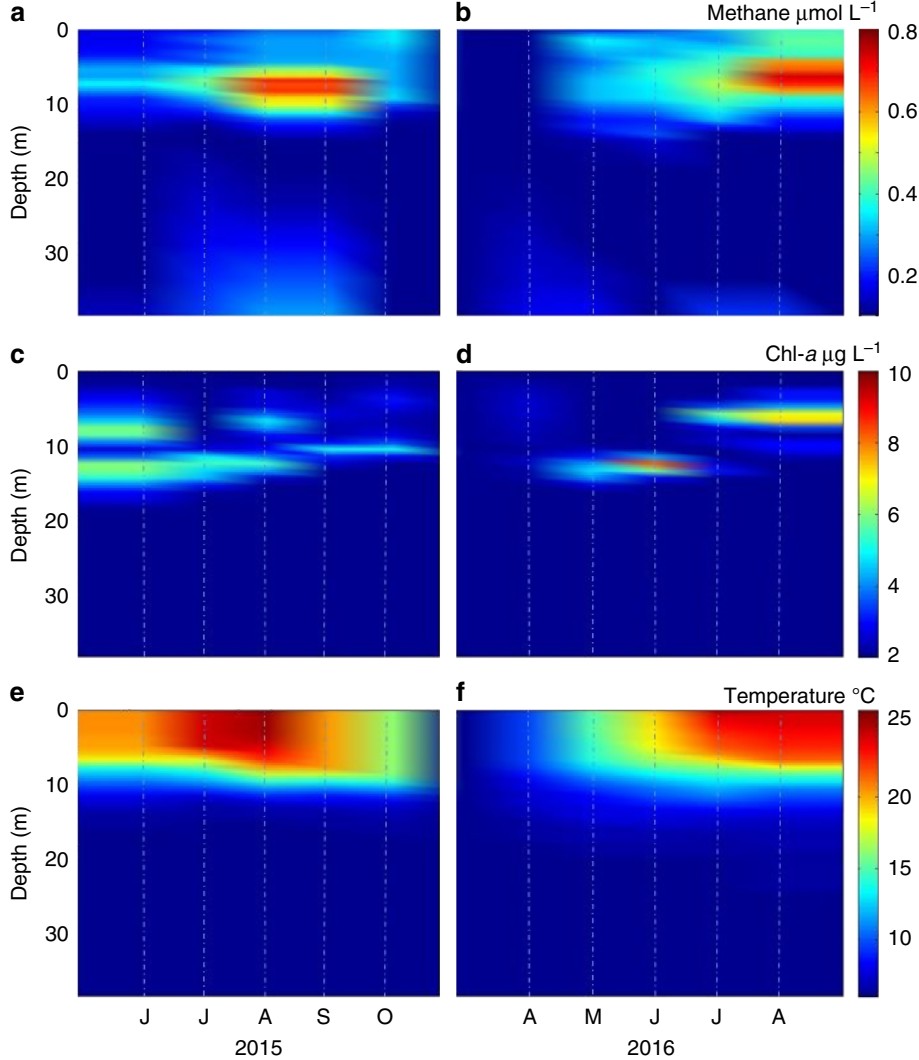

**Fig. 2** Evolution of dissolved CH$_4$ in the water column (0–40 m) of Lake Hallwil. Dissolved CH$_4$ (µmol L$^{-1}$) in **a** and **b**, chlorophyll a (Chl-a; µg L$^{-1}$) in **c** and **d** and temperature (°C) in **e** and **f** from June–October 2015 to April–August 2016 interpolated from measurements at the lake center (47°16.762 N, 8°12.791 E, St. A, Supplementary Fig. 1). Dashed vertical lines indicate the sampling date

presence of a double Chl-a maximum (Fig. 2c, d) is a recent phenomenon in Lake Hallwil that is particularly pronounced during the summer season. The presence of a surface chlorophyll maximum (SCM) in the epilimnion has been reported for several lakes[23]. In Lake Hallwil, while the SCM is associated with chrysophytes, chlorophytes, and diatoms, the deep chlorophyll maximum is associated with the filamentous cyanobacteria *Planktothrix rubescens*[24] (Supplementary Note 1).

Temperature-based basin scale vertical diffusivities $K_z$ (m$^2$ s$^{-1}$) were determined below 5 m depth by the heat budget method[25,26] (Methods). The profiles revealed how with the onset of stratification between May and August, water column stability ($N^2$) between 5 and 10 m increased from $1 \times 10^{-3}$ to $5 \times 10^{-3}$ s$^{-2}$, while at the same depths basin scale turbulent diffusion reached its minimum values ($\sim 1 \times 10^{-6}$ m$^2$ s$^{-1}$) (Supplementary Fig. 3).

**Methane heterogeneity.** Lateral (east–west) and longitudinal (north–south) heterogeneity of the CH$_4$ concentrations were investigated in 2015. A lateral transect ($\sim 1$ km east–west) of four CH$_4$ profiles was performed within a few hours at increasing distances from the shore toward the center (Fig. 3a). The longitudinal variability was also investigated with a south and center profile (Fig. 3b). The spatial variation of the CH$_4$ profile (defined

as the ratio of the standard deviation to the mean) is 50% smaller than the temporal variation of the profile performed at the lake center between June and August 2015. Therefore, given the spatial similarity of the metalimnetic CH$_4$ maximum and CH$_4$ concentrations in general, we considered the profile obtained at the center (as in Fig. 2a, b) representative for the entire lake production and transport dynamics.

**Oxidation rates.** Stable carbon isotopes of CH$_4$ ($\delta^{13}$C$_{CH4}$ values) were measured for in situ lake water incubations to investigate CH$_4$ oxidation. Seasonally, the CH$_4$ in the Lake Hallwil surface layer had an average $\delta^{13}$C$_{CH4}$ (June–August 2016) of $-60$‰ $\pm$ 2‰ ($n = 15$, all results reported in $\pm 1$ standard deviation, SD, unless otherwise indicated), and became isotopically enriched ($\sim -40$‰) below the CH$_4$ peak and thermocline (Fig. 4a, c, e). Incubations revealed a maximum CH$_4$ oxidation rate (MOx) of 6 nmol L$^{-1}$ d$^{-1}$ at 13 m depth (July–August 2016, Fig. 4d). While slight decreases of CH$_4$ concentrations were observed in water incubations above 10 m for the period July–August 2016 ($3.6 \pm$ 0.2 and $3.2 \pm 0.9$ nmol L$^{-1}$ d$^{-1}$ at 2 and 8 m depth, respectively), no change in isotope ratio was observed after 3 weeks of incubation (Fig. 4b, d).

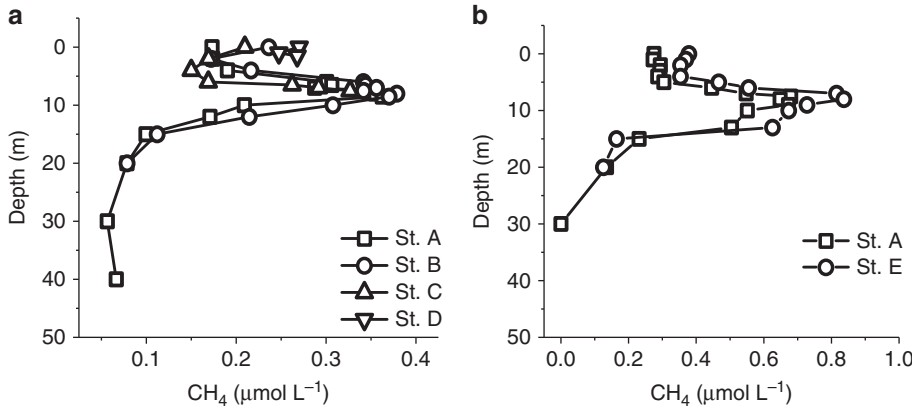

**Fig. 3** Dissolved $CH_4$ spatial heterogeneity in Lake Hallwil. Water column dissolved $CH_4$ profiles carried out within few hours on **a** 10 June 2015 from the east shore toward the center of the lake at stations A, B, C, D (45, 40, 20, 2 m lake depth, respectively) and **b** on 21 August 2015 from center (St. A) toward south (St. E, 20 m lake depth). See Supplementary Fig. 1 for sampling locations

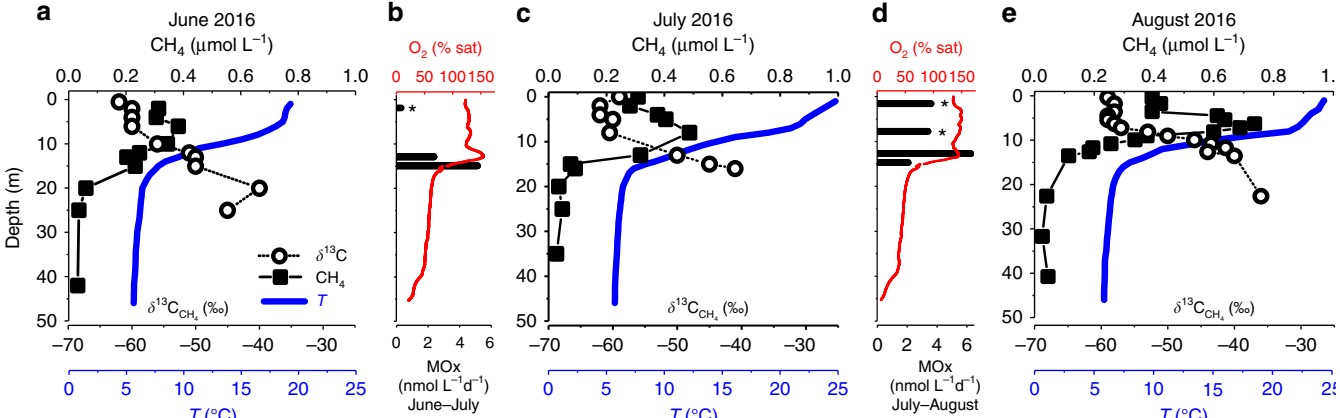

**Fig. 4** Water column $CH_4$ concentration and oxidation rates. **a**, **c**, **e** Water column profiles of $CH_4$, $\delta^{13}C_{CH4}$, and temperature for 17 June, 7 July, and 3 August 2016. $CH_4$ profiles show the distinctive metalimnetic maxima, while at 25–30 m concentrations were <0.05 μmol L$^{-1}$. The $\delta^{13}C_{CH4}$ profiles in the epilimnion are rather uniform with values around −62 to −60‰. **b**, **d** Average dissolved oxygen profiles and MOx rates obtained from in situ incubations for the periods June–July 2016 and July–August 2016 show maximum oxidation rates and correspondingly higher $\delta^{13}C_{CH4}$ values at 15 and 13 m depth, respectively. In the SML, oxidation rates were negligible for the period June–July 2016, and higher during the period July–August 2016 (3.6 ± 0.2 nmol L$^{-1}$ d$^{-1}$) although associated with no significant (2‰) change in $\delta^{13}C_{CH4}$ (*)

**Benthic fluxes.** Sediment porewater $CH_4$ concentrations in Lake Hallwil measured on cores retrieved at 3 and 7 m depth (Methods) averaged 1 mmol L$^{-1}$ at 5 cm b.s.s. (below sediment surface) with $\delta^{13}C_{CH4}$ of −68‰ and −66‰ in the upper 15 cm, respectively (Fig. 5). $CH_4$ diffusive flux at these depths was calculated with Fick's 1st law as 1.6 and 1.9 mmol m$^{-2}$ d$^{-1}$, respectively. $CH_4$ diffusive fluxes at 23 and 45 m depth were estimated as 5 and 6 mmol m$^{-2}$ d$^{-1}$, respectively (Fig. 5). The $\delta^{13}C$ values of porewater $CH_4$ at 23 and 45 m depth were about −75‰ (5 cm b.s.s.), which is 8‰ and 16‰ lower than those measured in littoral pore- and surface waters, respectively.

**Surface fluxes.** Surface $CH_4$ fluxes were measured with a floating chamber between June 2015 and August 2016. Average April–August 2016 evasion at the air–water interface corresponded to 0.6 ± 0.3 (1 SD, n = 28) mmol m$^{-2}$ d$^{-1}$, while surface $CH_4$ concentrations averaged 0.3 ± 0.1 (1 SD, n = 5) μmol L$^{-1}$. Measured $CH_4$ emission rates were compared with flux estimates based on wind speed for May–August 2016. Chamber-based $CH_4$ fluxes compared well with wind speed-derived diffusive fluxes

calculated according to the parameterization for a stratified water column by MacIntyre et al.[27] (0.8 ± 0.2 mmol m$^{-2}$ d$^{-1}$) (Supplementary Fig. 4). The flux estimate for negative buoyancy, typical for night convective mixing, is nearly double than what was estimated from chamber measurements as these were almost always taken during the day (Supplementary Fig. 4). Consequently, the surface flux component ($F_S$, Fig. 1) is a conservative estimate for the summer period.

**Mass balance.** To determine the overall $CH_4$ net production in the lake ($P_{net}$, Fig. 1), mass balances were performed within the two defined zones using the various rates shown in Tables 1 and 2. Sources and sinks during the lake stratified period (May–October) of 2016 were determined dividing the surface layer in two key zones as shown in Fig. 1. Zone 1, between 5 and 10 m depth, includes the steep thermocline where low (basin scale) turbulent diffusion dominates ($K_z = 1$–$2 \times 10^{-6}$ m$^2$ s$^{-1}$; Supplementary Fig. 3). Zone 2 is defined as the SML between 0 and 5 m depth where, on a seasonal scale, convection and wind-driven advective mixing dominates.

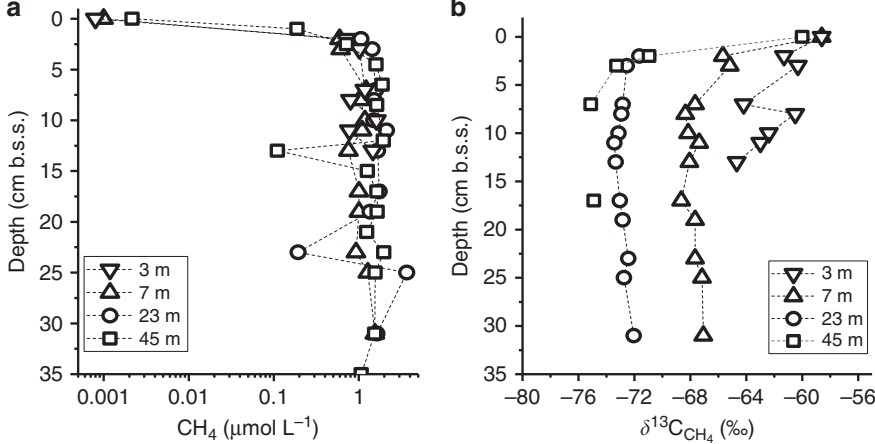

**Fig. 5** Porewater $CH_4$ concentrations and $\delta^{13}C_{CH4}$ values. **a** Profiles of porewater $CH_4$ concentrations (log scale) and **b** $\delta^{13}C$ values on sediment cores retrieved at deep sites (S3, 23 m and S4, 45 m) and at shallow stations (S1, 3 m and S2, 7 m) in the proximity of high lake surface $CH_4$ concentrations: 0.8 $\mu mol\,L^{-1}$ (-59‰) at the 3 m site and 1 $\mu mol\,L^{-1}$ (-59‰) at the 7 m site. The 3 m site showed extensive gas voids, and some gas bubble release in the retrieved core. All profiles were performed on September 2016 except for S4, which refers to June 2015

**Metalimnion mass balance**. Higher $CH_4$ concentrations in the metalimnion diffuse toward the lower concentrations down in the hypolimnion and up to the SML. The vertical transport of dissolved $CH_4$ from the metalimnion to the SML and hypolimnion is driven via turbulent diffusivity and the concentration gradients, where the basin scale diffusivity, $K_z$, was determined to be $\sim 1 \times 10^{-6}\,m^2\,s^{-1}$. The average vertical $CH_4$ diffusion ($F_Z$) was determined by Fick's 1st law (Eq. 1) as:

$$F_z = -K_z \frac{\partial C}{\partial z}; \quad \left[ mmol\,m^{-2}d^{-1} \right] \qquad (1)$$

where $C$ determines the $CH_4$ concentration and $z$ the depth. The vertical flux was calculated to be $\sim 14\,nmol\,L^{-1}\,d^{-1}$ (0.07 mmol m$^{-2}$ d$^{-1}$) both upward and downward from the peak that formed between June and August 2016.

Such small $CH_4$ fluxes through the metalimnion are caused by low turbulent diffusivities (Supplementary Fig. 3). However, horizontal transport at the thermocline can be several orders of magnitude higher than vertical diffusion[25]. We therefore consider a lateral transport from the littoral sediment in the mass balance ($F_L = \sim 9\,nmol\,L^{-1}\,d^{-1}$, see Methods and Fig. 1), with the assumption that the added mass is well-mixed horizontally across the lake over the time scale of the calculations. Bubble transport of bottom water methane facilitated by the aeration system was also considered (Discussion) as a potential contribution to $CH_4$ concentrations in zone 1 ($F_D = \sim 3\,nmol\,L^{-1}\,d^{-1}$, see Methods and Fig. 1).

With Fick's 2nd law, we determined the depth-dependent $CH_4$ production ($P_{gross,m}$) in zone 1 expressed by the sum of losses by diffusion ($F_z$, Fig. 1) and oxidation and inputs from the littoral zone and from the hypolimnion ($F_L$ and $F_D$; Fig. 1),

$$\frac{\partial C}{\partial t} = K_z \frac{\partial^2 C}{\partial z^2} + P_{gross,m}; \quad \left[ nmol\,L^{-1}d^{-1} \right] \qquad (2)$$

with $t$ as time. Equation 2 can be applied in both the sediment and the stratified water column[28]. Although only applicable below the SML, where the water column is stably stratified, this approach presents the advantage of a direct estimation of system-wide production rates ($P_{gross,m}$) with high vertical resolution.

Local methane production ($P_{net,m}$) was calculated by removing the estimated littoral contributions as $P_{gross,m} - (F_L + F_D)$ (Fig. 1), for both periods June–July and July–August 2016 (Fig. 6a, b), indicating an average (June–August) aerobic methane production

($P_{net,m}$) of $\sim 5.0 \pm 5.0\,nmol\,L^{-1}\,d^{-1}$ between 6 and 7 m. Yet, when net $P$ rates are integrated over the metalimnion (zone 1 in Fig. 6), $P_{net,m}$ become negligible at $0.3 \pm 3.0\,nmol\,L^{-1}\,d^{-1}$ (Table 1).

Net production rates for both analyzed periods indicate that below 9 m, $CH_4$ is consumed due to aerobic oxidation (MOx) at rates between 5.0 and $20\,nmol\,L^{-1}\,d^{-1}$ (Fig. 6a, b). These estimates are in good agreement, although slightly greater than MOx rates obtained by in situ incubations at 13 and 15 m ($\sim 6.0\,nmol\,L^{-1}\,d^{-1}$, Fig. 4b, d). However, when $CH_4$ diffusion to the water column due to ebullitive inputs from sediments below 10 m depth is considered negligible[29], the obtained consumption rate ($P_{net,m}$) between 10 and 15 m is 25% lower, thus closer to the oxidation rates from incubation experiments.

**Surface mass balance**. Surface $CH_4$ fluxes ($0.6 \pm 0.3\,mmol\,m^{-2}\,d^{-1}$, Supplementary Fig. 4) and concentrations ($0.3 \pm 0.1\,\mu mol\,L^{-1}$, Fig. 2a, b) exhibit relative temporal uniformity in contrast to the eight-fold $CH_4$ accumulation between 5 and 10 m depth observed at the lake center throughout the same time period. Between June and August (both 2015 and 2016), most of the $CH_4$ surface flux originates from the relatively well-mixed top 5 m (Fig. 2). Therefore, we assume on the seasonal scale that the surface layer can be modeled as a well-mixed reactor, and $CH_4$ net production rates ($P_{net,s}$) can be estimated as follows:

$$\frac{\partial C}{\partial t} \forall = (Q_R C_R) + A_s F_L + A_p F_Z + P_{net,s} \forall \\ - (MOx \forall + A_p F_S); \quad \left[ mol\,d^{-1} \right] \qquad (3)$$

where $A_s$ and $A_p$ are the sediment surface (between 0 and 5 m) and the lake planar area, respectively. Contributions to the $CH_4$ budget in the SML (zone 2), as shown in Fig. 1, are listed in Table 2 (see Methods for each term calculation).

As surface concentrations do not vary much seasonally (Fig. 4a, c, d), we assume steady state ($\frac{\partial C}{\partial t} \forall = 0$) and solve the mass balance in the top 5 m revealing a source of $CH_4$ ($P_{net,s}$ in Eq. 3) of $110 \pm 60\,nmol\,L^{-1}\,d^{-1}$ ($73 \pm 40\,kg$ of $CH_4$ per day during stratified periods). This rate is about a 100× higher than in the metalimnion and accounts for up to 90% of total measured $CH_4$ evasion (Table 2). This is a conservative estimate, as it was assumed that the whole sediment surface of the lake from 0 to 5 m is subject to highest rates of ebullition ($1.2 \pm 0.8\,mmol\,CH_4\,m^{-2}\,d^{-1}$ dissolved in water, representing 1.2% of the lake area according to Flury et al.[30]). The mass

**Table 1 Methane mass balance for the metalimnion**

| $CH_4$ flux | Description | mol $d^{-1} \pm 1$ SD | kg $d^{-1}$ | nmol $L^{-1} d^{-1}$ |
|---|---|---|---|---|
| $F_L$ (eb) | Dissolution from littoral ebullition | $162 \pm 108$ | $2 \pm 1$ | $4.0 \pm 2.5$ |
| $F_L$ (sed) | Diffusion from littoral sediments | $240 \pm 28$ | $4.0 \pm 0.5$ | $6.0 \pm 0.6$ |
| $F_D$ | $CH_4$ contributions from the aeration system | 0–120 | 0–2 | 0–3 |
| $P_{gross,m}$ | Including production from Eq. 2, results from losses: MOx and $F_Z$, and transport: $F_L$ and $F_D$ | $400 \pm 40$ | $6.0 \pm 0.5$ | $10 \pm 1$ |
| $P_{net,m}$ | Net $CH_4$ production | $14 \pm 118$ | $0.2 \pm 2$ | $0.3 \pm 3$ |

Methane mass balance components (mean $\pm 1$ SD or range) and net production for the metalimnion (zone 1; 6–10 m depth) calculated as $P_{net,m} = P_{gross,m} - (F_L + F_D)$

balance includes MOx rates from in situ incubations (Table 2), although these are negligible compared to the surface losses of $CH_4$ to the atmosphere. While our uncertainty analysis detailed in Supplementary Table 1 indicates that the error associated with $P_{net,s}$ ($110 \pm 60$ nmol $L^{-1} d^{-1}$) is largely due to air–water exchange estimates that, as described above, are conservative.

**Methane sources from isotope evaluation**. To assess possible similarity between water column and porewater $CH_4$ formation, we investigate the difference between the isotope measurements of $CH_4$ and methanogenic precursors (total and dissolved organic carbon, TOC and DOC). Based on calculations according to Bogard et al.[9] (Methods), we infer a smaller difference between the isotope values of carbon source and $CH_4$ produced in the oxic water column (−32 to −29‰) as compared to the sediment methanogenesis (−44 to −41‰, Table 3).

The apparent fractionation factor ($\alpha_{app}$) during methanogenesis was defined as in Conrad et al.[31] where the isotopic signature of source $CH_4$ was estimated by correcting the $\delta^{13}C_{CH4}$ ambient measurement for the isotopic fractionation due to diffusion and oxidation (Methods). Sediment $CH_4$ production of Lake Hallwil exhibits an $\alpha_{app}$ of 1.056–1.060, which is characteristic for environments dominated by acetate-dependent methanogenesis. Estimates for the water column SML methane production show a smaller fractionation factor ($\alpha_{app} = 1.045$). Consequently, in Lake Hallwil we observed a different isotopic fractionation between the $CH_4$ produced in sediments and in the SML, however both characteristic for acetoclastic methanogenesis.

**Discussion**

This study quantifies $CH_4$ production rates in the oxic surface layer of a mesotrophic Swiss lake by estimating the system-wide $CH_4$ transport, dynamics, and emissions. The conservative mass balance performed for this study illustrates that, during periods of lake stratification (April–October), up to 90% of the $CH_4$ that is emitted to the atmosphere ($73 \pm 40$ kg $d^{-1}$ or $26 \pm 14$ t $y^{-1}$) is the product of unknown production process(es) that primarily occur in the SML (top 5 m) of Lake Hallwil. The metalimnion $CH_4$ concentration maximum, often observed in mesotrophic lakes, does not correspond to a maximum production rate. The observed metalimnetic $CH_4$ production rate ($P_{gross,m}$) of about 10 nmol $L^{-1} d^{-1}$ can be largely explained by lateral transport from the adjacent sediments (Table 1). Negative production rates below 10 m (Fig. 6) are explained by oxidation of $CH_4$ as confirmed by in situ bottle incubations (~6 nmol $L^{-1} d^{-1}$), where a $CH_4$ stable carbon isotope ratio increase of 20‰ was observed after 1 month for both periods June–July and July–August (Fig. 4b, d).

In the SML of Lake Hallwil, the situation is vastly different. We show that during the stratified season, the most significant production rate ($P_{net,s} = 110 \pm 60$ nmol $L^{-1} d^{-1}$) is mostly expressed in these upper 5 m and not in the metalimnion. Bottle incubations in the SML show either negligible $CH_4$ oxidation (0.3 nmol $L^{-1} d^{-1}$, June–July 2016) or higher oxidation ($3.6 \pm 0.2$ nmol $L^{-1} d^{-1}$, July–August 2016) with negligible change in isotope values

(2‰ after 1-month incubation) (Fig. 4b, d). This may indicate oxidation is compensated by a $CH_4$ production mechanism. The magnitude of the surface $CH_4$ production is however masked by the relatively rapid water–air exchange. As a result of the $CH_4$ loss to the atmosphere, the observed $CH_4$ concentrations remain lower and fairly consistent in the surface layer vs. the metalimnion.

Surface water $CH_4$ oversaturation has been suggested to be produced in situ under oxic conditions[10,11]. Current hypotheses for lakes are derived from the strong correlations observed between OMP ($P_{net,s}$), photosynthesis and $O_2$ concentration[4,13]. However, the characteristic $CH_4$ peak may lead to misinterpretations when seeking correlations. Photosynthesis and $O_2$ concentration are positively correlated to autotrophic biomass, whose distribution in the water column is strongly related to the physical water column structure[32]. That is, the variables listed above also tend to correlate with water column stability. Comparing $CH_4$ to Chl-a, turbidity and water column stability ($N^2$) revealed that the $CH_4$ concentration only correlates significantly with $N^2$ (Supplementary Table 2), suggesting a physical component behind the observed $CH_4$ accumulation in the metalimnion. This supports that the highest production rates are expressed at the ventilated surface layer, while the $CH_4$ in the metalimnion represents only a local accumulation that supplies very little $CH_4$ to the surface layer. Therefore, relying on correlations of $CH_4$ concentration with other variables alone is misleading, while using the production rates for correlations provides a clearer picture of each vertical zone's importance in sustaining $CH_4$ emissions.

Our approach reveals for the first time that $CH_4$ production in oxic waters ($P_{net,s}$, Fig. 1) appears to decrease rapidly with water depth, where production rates in the top 5 m are 100 times greater than in the metalimnion (Fig. 7). Our mass balance-based production estimate of $110 \pm 60$ nmol $L^{-1} d^{-1}$ is remarkably close to the OMP rates observed by laboratory incubations (50 nmol $L^{-1} d^{-1}$)[13] and lake enclosures (~200 nmol $L^{-1} d^{-1}$)[9]. Despite the different approaches, methane production rates lay within a surprisingly narrow range. Thus our results both support the growing body of evidence for OMP as well as better constrains the rates now reported in multiple freshwater environments.

Most studies on $CH_4$ oxidation are carried out in stratified eutrophic systems, where $CH_4$ concentrations at the thermocline are higher than in Lake Hallwil (up to 600 time higher[20]). In such cases, the corresponding MOx rates can be about 100 times greater than what is found in Hallwil (e.g., 1 µmol $L^{-1} d^{-1}$). Our MOx results are however in good agreement with similar mesotrophic systems (e.g., Lake Biwa; ~5 nmol $L^{-1} d^{-1}$ from dark incubations of 15 m deep water[19]) and are in the order of the lower range measured by Bogard et al.[9] for Lake Cromwell (60 nmol $L^{-1} d^{-1}$). Methane oxidizing bacteria favor isotopically lighter $CH_4$, leaving a residual $CH_4$ with a higher $\delta^{13}C_{CH4}$ value. Both high oxygen concentrations and light exposure have been shown to significantly inhibit MOx[19,33,34]. In Lake Hallwil, these

**Table 2 Methane mass balance for the surface mixed layer**

| $CH_4$ flux | Description | mol d$^{-1}$ ± 1 SD | kg d$^{-1}$ | nmol L$^{-1}$ d$^{-1}$ |
|---|---|---|---|---|
| $F_S$ | Evasion from surface | 5040 ± 2520 | 80 ± 40 | 121 ± 60 |
| MOx | Methane oxidation | 150 ± 8 | 2 ± 0.1 | 3.6 ± 0.2 |
| $F_L$ (eb) | Dissolution from littoral ebullition | 134 ± 89 | 2 ± 1 | 3 ± 2 |
| $F_L$ (sed) | Diffusion from littoral sediments | 196 ± 22 | 3 ± 0.3 | 5 ± 0.5 |
| $F_R$ | Input from rivers | 0–207 | 0–3 | 0–5 |
| $F_Z$ | Diffusion from metalimnion | 252 ± 84 | 4 ± 1 | 6 ± 2 |
| $P_{net,s}$ | Net $CH_4$ production | 4600 ± 2500 | 73 ± 40 | 110 ± 60 |

Methane mass balance components and relative flux rates for the surface mixed layer (zone 2; 0–5 m depth). Note the addition of $Q_R \times C_R$ ($=F_R$) as the contribution of river input to the methane pool. Net $CH_4$ production is calculated as $P_{net,s} = (A_P F_S + MOx) - (F_R + A_s F_L + A_P F_Z)$

findings are supported by negligible oxidation rates measured in situ (June–July 2016) and by the lighter $\delta^{13}C_{CH4}$ values above 6 m (Fig. 7). July–August surface water incubations showed a slight $CH_4$ decrease with no isotope values change. With a $\delta^{13}C_{CH4}$ equal to −62‰ at Time 1, as from in situ measurements, the isotope value associated to the measured consumption rates of 3 nmol L$^{-1}$ d$^{-1}$ should have been in the order of −41‰ instead of the final measured −64‰ (assuming MOx with fractionation factor of 20‰). This might suggest a local compensation with production of isotopically lighter $CH_4$.

$\delta^{13}C_{CH4}$ values at the surface (−62 to −60‰) are on average lower than those reported in other OMP studies (−50‰, Lake Stechlin[3]; −55‰, Lake Lugano[4]; −40‰, Lake Cromwell[9]; −62 to −21‰, Lake Biwa[5]) but are 5 and 15‰ higher than the measured porewater $\delta^{13}C_{CH4}$ at 7 and 45 m lake depth (−65‰ and −75‰, respectively). Interestingly, similar differences between lake surface and porewater $CH_4$ stable carbon isotopes were reported in other meso-oligotrophic lake studies[3–5]. Biogenic methanogenesis of freshwater systems is known to be strictly anaerobic and mainly based on the fermentation of acetate, the most favorable substrate for freshwater methanogens[35]. The fractionation factor $\alpha$ during acetoclastic $CH_4$ formation can vary between 1.009 and 1.065[36–39]. Yet the carbon isotope composition of $CH_4$ can be influenced by the type of acetate precursor[40], the production mechanism(s) and pathways, and relatively little is known about methanogenesis of oligotrophic lake surface sediments[41]. Lake Hallwil's apparent fractionation (1.056–1.060) of sediment $CH_4$ production indicates an acetate-dependent methanogenesis, which is in good agreement with temperate, oligotrophic lake sediments (e.g., 1.065, Lake Stechlin[41]).

While isotopic studies of OMP have been reported for plant-derived organic materials exposed to ultraviolet light[42], virtually nothing is known about the stable carbon isotope fractionation associated to aerobic $CH_4$ production in aquatic systems. Our estimates for the water column SML $CH_4$ production show a smaller fractionation factor compared to the sediments ($\alpha_{app}$ = 1.045), but still characteristic for acetoclastic methanogenesis and similar to what was found by Bogard et al.[9] for their enclosure experiments ($\alpha_{app}$ = 1.02–1.04). Furthermore, we assessed a smaller difference between isotope values of precursors (DOC, TOC) and product ($CH_4$) from water column methanogenesis (32–29‰) as compared to the sediment methanogenesis (44–41‰, Table 3), which may indicate that water column $P_{net}$ derives from a distinct pathway, not linked to the sediments. However, the estimates of fractionation factors performed here, as in the majority of methanogenesis studies, are based on the assumption that there is no major methanogenic precursor other than acetate or $CO_2$[31]. Arguably, additional and perhaps novel pathways should be evaluated.

Only recently, MPn biodegradation has been indicated as a possible source of $CH_4$ in oceans[43], a theory which was confirmed to apply to mesotrophic lakes in recent work on Lake Yellowstone[15]. However, in the mentioned study, and contrary to our findings, laboratory efforts focused on samples taken at the metalimnetic $CH_4$ peak that occurs during stratification. MPn can derive from anthropogenic activity (e.g., herbicide glyphosate) and is known to contribute to the phosphonate pool in lakes and their watersheds[44]. It was furthermore shown that, when environmentally limited, phosphate can be regenerated from semi-labile dissolved organic matter through the C-P lyase pathway with formation of $CH_4$[45]. Indeed, the expression of the C-P lyase gene found in many freshwater cyanobacteria[46,47] is induced by $P$ limitation[48]. This hypothesis suits Lake Hallwil mainly for two reasons: low water column $P$ concentrations (3 µg L$^{-1}$ in top 5 m, DIN:$P_{tot}$ > 100 in August 2015), and surface (1 m) $CH_4$ concentrations that correlate with DOC (Supplementary Fig. 5a). Lake Hallwil is surrounded by an intense agricultural landscape which is potentially a source for MPn, although the absence of strong lateral $CH_4$ gradients points toward the relationship with DOC (Supplementary Fig. 5b) supporting a link between the oxic water $CH_4$ production and algal-derived organic matter substrate availability[9].

An additional explanation for OMP could be the breakdown of chromophoric dissolved organic matter by solar radiation in the ultraviolet and visible range[49,50] to organic compounds that serve as precursors for non-microbial $CH_4$ production[51,52]. The photolysis of organic matter was shown to supply $CH_4$ to the surface waters at relatively low rates in Saguenay River ($4.36 \times 10^{-6}$ mol m$^{-2}$ yr$^{-1}$)[53]. Such a process would directly relate to the trophic state (i.e., clarity)[54]. In Lake Hallwil, this seems supported by the light penetration/$CH_4$ production curve relationship (Fig. 7; Supplementary Fig. 5a). While this aspect needs further investigation, we conclude that $CH_4$ production is occurring in the SML, regardless of the source/process.

Littoral sediments are known to contribute to $CH_4$ emissions via ebullition[30]. The dissolution of the rising bubbles[55] and enhanced sediment $CH_4$ diffusion from gas-charged sediments[56] could contribute to the high dissolved $CH_4$ concentrations in the littoral zone. Thus $CH_4$ release from the littoral sediments and subsequent horizontal transport could be another source of $CH_4$ in lake surface waters[8], however these contributions are less important in the SML than at increasing depths. In Lake Hallwil's metalimnion, we assess that lateral transport accounts for 10 ± 1 nmol $CH_4$ L$^{-1}$ d$^{-1}$ (Table 1) leading to an accumulation of ~5 ± 5 nmol L$^{-1}$ d$^{-1}$ (Fig. 6) of which an average of 0.3 ± 3 nmol L$^{-1}$ d$^{-1}$ is locally produced/consumed. Contrarily, in the SML we estimate a significant and unaccounted for internal source of $CH_4$ ($P_{net,s}$, Fig. 1) of 110 ± 60 nmol L$^{-1}$ d$^{-1}$ that is in the same range of what was estimated by laboratory and mesocosm-based studies[9,13].

Low $CH_4$ concentrations (<0.05 µmol L$^{-1}$) between 25 and 45 m depth at the lake center led to the conclusion that any $CH_4$ diffusing from deep sediments (5 and 6 mmol m$^{-2}$ d$^{-1}$ at 23 and 45 m, respectively) does not reach the metalimnion or the SML.

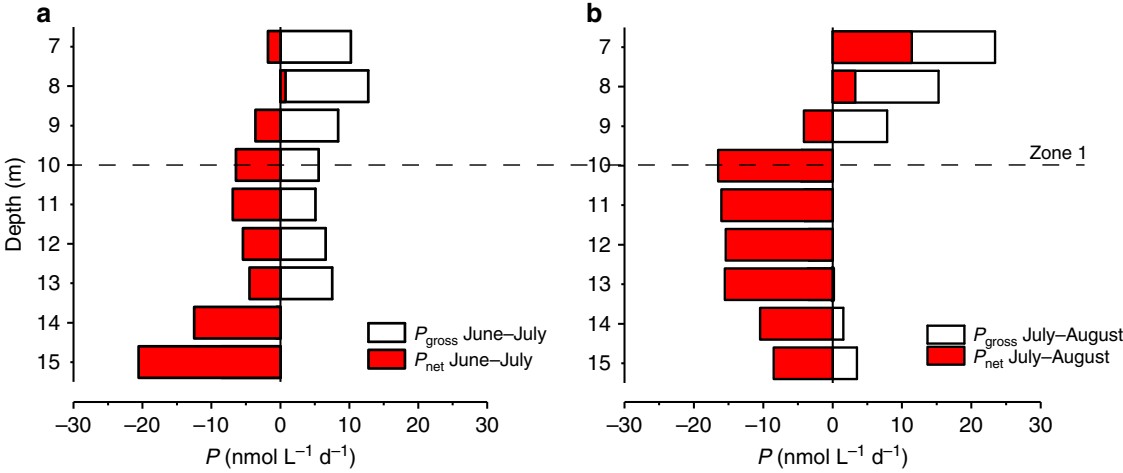

**Fig. 6** High-resolution water column $CH_4$ production and consumption rates. System-wide $CH_4$ production rates (white bars, $P_{gross,m}$) obtained by Fick's 2nd law and local production $P_{net,m}$ (red bars) = $[P_{gross,m} - (F_L + F_D)]$ for the period **a** June–July and **b** July–August 2016. $P_{net,m}$ rates indicate an overall net $CH_4$ production/consumption of $-0.3 \pm 3.0$ nmol $L^{-1}$ $d^{-1}$ in the metalimnion (zone 1; 6–10 m), while consumption rates increase up to $-20$ nmol $L^{-1}$ $d^{-1}$ below 10 m. This analysis shows that a small net $CH_4$ production (red bars) is observed between 7 and 8 m between June and August 2016 ($\sim 5 \pm 5$ nmol $L^{-1}$ $d^{-1}$). Note that the depth integrated production starts between 6 and 7 m; above 6 m depth (SML) the vertical diffusion ($K_z$) used in Fick's 2nd law cannot be accurately inferred with the heat budget method (Methods). The dashed line indicates the lower limit of zone 1

The highest oxidation rates are likely taking place within the surface sediments as reported for other studies in mesotrophic lakes[5,57]. However, the presence of the aeration system may favor the transport of bottom water $CH_4$ within rising bubbles. This contribution to the bulk $CH_4$ content at 5–10 m depth was quantified using modeling of air bubbles at equilibrium with the highest measured bottom water (46.5 m) $CH_4$ concentration (7 $\mu$mol $L^{-1}$, Methods). The maximum input to the thermocline (zone 1) was estimated as 120 mol $d^{-1}$, which represents only 3–6% of the estimated $P_{net,s}$. However, even such contribution to the mass balance is very conservative. In fact, if the plume was transporting $CH_4$ from the benthic boundary layer upward, then we would see elevated concentrations below the thermocline in the area of plume detrainment (the main sampling station A, on Supplementary Fig. 1, is only ~250 m south of the aeration system diffuser ring). Here the profiles show that dissolved $CH_4$ below the thermocline to ~40 m depth was near the detection level of the method in all cases (Fig. 4a, c, e), indicating the concentrations within the plume itself are likely near this background concentration.

During summer stratification, the SML is generally restricted to the top several meters of the lake, effectively isolating the surface $CH_4$ from oxidation processes[58]. Therefore, $CH_4$ formed in the surface layer can be continuously and rapidly delivered to the atmosphere. Consequently, longer stratification periods from a warmer climate could result in longer periods of OMP-related $CH_4$ evasion ($P_{net,s}$, Fig. 1). Similar conclusions were drawn for the marine environment, for which aerobic $CH_4$ production is suggested to be sensitive to changes in water column stratification and P limitation[10].

In the present study, we used detailed whole-lake mass balancing combined with incubation and isotopic approaches to show that in Lake Hallwil, during the stratified period, up to 90% of the emissions ($26 \pm 14$ tons per year) result from surface layer $CH_4$ production. The estimated production rates are in agreement with what is suggested by other laboratory and mesocosm-based studies[9,13]. However, with our whole-lake approach, this is the first study to determine that the highest production rates occur within the lake SML rather than within the often suggested metalimnion, and are depth-correlated with DOC and light penetration. Several oligo- and mesotrophic lakes such as Lake

Stechlin[13], Lake Lugano[4], Lake Matano[45], Lake Yellowstone[15], and Lake Geneva (Supplementary Fig. 6) have been recently studied and reported the occurrence of OMP. The present findings for Lake Hallwil frame an important and underestimated contribution to atmospheric $CH_4$, as oligo-mesotrophic systems are typically not considered as significant greenhouse gas sources.

Consequently, attention should be paid to the result of restoration programs (deeper light penetration, low phosphorous), which could indirectly lead to enhanced greenhouse gas emissions—another paradox concerning aquatic systems that has been so far overlooked.

## Methods

**Study site.** Lake Hallwil (Canton of Argovia, Switzerland) is a mesotrophic lake with a surface area of 10.2 km², a mean depth of 28.6 m, and a maximum depth of 46.5 m. The basin water volume is 0.29 km³ with negligible riverine inflow (2.5 m³ s⁻¹), of which 50% flows in from upstream Lake Baldegg in the south. Dominant winds are from the west resulting in limited large-scale seasonal mixing of the north–south-oriented lake sheltered by hills[24]. Since 1986, Lake Hallwil has had no ice cover in winter. A restoration process was aided since autumn 1985 by the installation of a bubble plume hypolimnetic aeration system designed to prevent a complete loss of oxygen in deep water[22] (Supplementary Note 1).

**Limnological measurements.** Monthly water column profiles at station A (45 m lake depth, 47°16.762 N, 8°12.791 E, Supplementary Fig. 1) were conducted with a multiparametric probe (6600 V2, YSI, USA until March 2016 and EXO2, YSI, USA afterward) equipped with temperature, conductivity, Chl-a, and turbidity sensors. Additionally, Secchi depth ($Z_s$), concentrations of DOC, and total phosphorous (October, May, August) were measured (using standard methods; www.labeaux.ch) and provided by the Department of Civil Engineering, Transportation and Environment of the Canton of Argovia, Switzerland.

Values for buoyancy frequency ($N^2$) were calculated from temperature, salinity, and pressure data as:

$$N^2 = -g\left(\frac{1}{\rho}\frac{\partial \rho}{\partial z} - \frac{g}{c^2}\right); \; [s^{-2}] \quad (4)$$

where $\rho$, $g$, and $c$ are the density, earth's gravitational acceleration, and speed of sound, respectively. The fraction of light ($I$) penetrating at depth $z$ ($I_z/I_0$) for June and August 2016 was calculated by the Lamber Beer equation:

$$I_z/I_0 = e^{-kz}; \; [-] \quad (5)$$

where the extinction coefficient $k$ was inferred as in Wetzel et al.[59] by measured Secchi depth (1.7/$Z_s$).

**Table 3 Evaluation of stable carbon isotope fractionation in porewater and surface water**

|  | $\delta^{13}C_{CH4}$ | $\delta^{13}C_{CO2}$ | $\delta^{13}C_{DOC}$ | $\delta^{13}C_{TOC}$ | $\delta^{13}C_{DOC,TOC}$–$\delta^{13}C_{CH4}$ |
|---|---|---|---|---|---|
| Sediment/pw (St. D, 15 cm b.s.s.) | −70 to −65‰ | −13‰ | −28‰ | −28‰ | −42 to −37‰ |
| Sediment/pw (St. A, 15 cm b.s.s.) | −75 to −72‰ | −16‰ | −31‰ | −34‰ | −44 to −41‰ |
| SML (St. A, 0.5 m) | −60‰ | −17‰ | −28‰ to 31‰ | −28‰ to 31‰ | −32 to −29‰ |

Stable carbon isotope ratio of $CH_4$ ($\delta^{13}C_{CH4}$) and its possible carbon precursors (TOC, DOC) measured in Lake Hallwil sediment and porewater (pw) at St. D (2.5 m deep) and St. A (45 m deep) and surface mixed layer (SML, St. A, lake center). Measurements refer to June–August 2015. Values in italics are assumed from literature[68] (see Supplementary Fig. 1 for stations map)

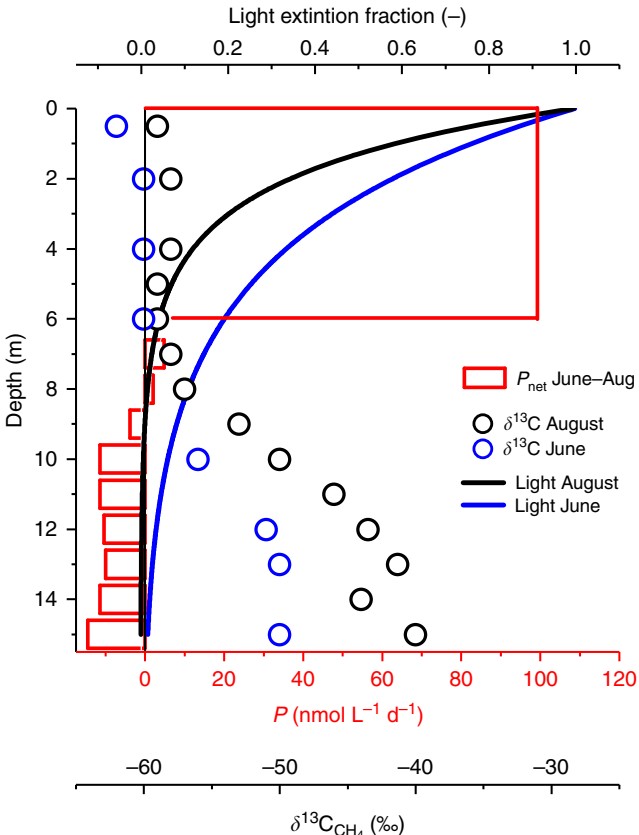

**Fig. 7** Linking water column $CH_4$ production oxidation and light extinction. Net $CH_4$ production in the top 5 m as derived from mass balance (Eq. 3) and below 6 m depth from Fick's 2nd law (Eq. 2). Light extinction curves were based on Secchi depths ($Z_s$) of 6 m in June (blue line) and 3.5 m in August (black line). The profile of $\delta^{13}C_{CH4}$ is added to illustrate the relationship between light extinction and $CH_4$ oxidation

**Water column $CH_4$ and $CO_2$ concentration, and $\delta^{13}C$.** Monthly $CH_4$ concentration profiles were taken at the deep Station A between June–October 2015 and April–August 2016. A 1 km long west–east transect was performed on 11 June 2015 at Stations A, B, C, D (45, 40, 20, and 2 m lake depth, Supplementary Fig. 1). A 3 km longitudinal transect (center–south) was performed on 21 August 2015 at Stations A and E (45 and 20 m depth, Supplementary Fig. 1).

For dissolved $CH_4$ concentration, water samples were obtained with a 5-L Niskin bottle at a maximum depth resolution of 0.5 m, where the metalimnetic $CH_4$ peak was expected.

Until October 2015, water samples for $CH_4$ concentration were collected with 60 mL syringes (Plastipak, Becton-Dickinson). One depth at the time, 20 mL of the syringe volume was replaced with ambient air ($CH_4$ = 1.75 ppm) and equilibrated by vigorously shaking for at least 2 min. The 20 mL gas volume was preserved in serum bottles prefilled with $CH_4$ free saturated NaCl solution and capped with gas tight butyl rubber stoppers (GMT Inc., USA). The gas sample headspace was created by injecting the gas volume (20 mL) into the serum bottles with a needle (0.6 × 30 mm, 23 G) and simultaneously evacuating the same volume of NaCl solution through a second needle previously inserted in the septum. The headspace was analyzed on the same day on a portable greenhouse gas (GHG) analyzer (UGGA; Los Gatos Research, Inc., USA).

From April 2016 on, dissolved concentrations of both $CO_2$, $CH_4$, and their stable C isotope ratio were measured. Therefore, samples were gently transferred from the Niskin sampler into a 1 L glass bottle (Duran GmbH, Mainz, Germany) overflowing to replace the volume about three times. A headspace was made immediately by replacing 400 mL of the sampled water via a 2-way stopcock valve with ambient air. With the valves closed, the bottle was shaken vigorously for 2 min. The headspace was then transferred into a 1 L gas-sampling bag (Supel Inert Multi-Layer Foil) via a 2-way stopcock by gently injecting lake water back into the bottom of the bottle. The gas samples were measured within 1 day on a Cavity Ring-Down Spectrometer analyzer (Picarro G2201-i, Santa Clara, CA, USA) for immediate reading of concentrations in the gas phase (ppm) and stable isotope ratio ($\delta^{13}C$ in ‰ vs. VPDB standard). Instrument-specific precision at ambient concentrations ($1-\sigma$ of 5 min average) is <0.16‰ for $\delta^{13}C$ of $CO_2$, <1.15‰ for $\delta^{13}C$ of $CH_4$, for $[^{12}CO_2]$ is 200 ppb + 0.05% of reading and for $[^{12}CH_4]$ is 5 ppb + 0.05% of reading. Water $CH_4$ and $CO_2$ concentrations were back calculated according to Wiesenburg and Guinasso[60] accounting for water temperature, air concentration (assuming 1.75 and 410 ppm for $CH_4$ and $CO_2$ respectively), and the resulting headspace/water ratio in the bottle.

Each sample procedure from the Niskin bottle to the gas bag takes ~10 min. To prioritize a higher vertical resolution, given the time consuming procedure (a longer time would increase the uncertainty linked to the natural variability of the water column structure), we performed replicates only once and applied the determined coefficient of variation (averaging 10.0% for $CH_4$ and 10.6% for $CO_2$ over a set of eight replicates) to the data set as a ±range of uncertainty of the measurement.

**$CH_4$ oxidation rates.** During June–August 2016, net methane oxidation rates were measured by incubating three replicates of 120 mL water from 2, 8, 13, 15, 25, and 35 m at Station M1 (Supplementary Fig. 1). The 120 mL glass bottles were previously soaked in 3 N HNO and rinsed with ultrapure water (MilliQ). The water samples from each depth were collected with a Niskin bottle sampler and immediately transferred to the incubation bottles while letting 3–4 volumes overflow prior to crimp capping the bottle headspace free. After sealing, incubations were started as soon as possible at in situ conditions by hanging the bottles along the thermistor chain at their corresponding sampling depth. Because dilutions were not made and nutrients were not added, there was no reason to expect any lag in the activity of the enclosed bacterial populations. Methane concentration was determined after 23 days in July and after 27 days in August on each of the bottles by replacing 40 mL with synthetic air ($CO_2$- and $CH_4$-free, Carbagas AG 2011) and analyzing the headspace on a Picarro analyser after complete equilibration by vigorous shaking (>2 min). The net rate of $CH_4$ oxidation was calculated by the decrease from in situ concentrations at the time when incubation started. To verify that the remaining $CH_4$ was an oxidized residue, we applied an isotope fractionation ($\varepsilon$) of 20‰[61] to calculate isotope composition of $CH_4$ in a Rayleigh fractionation model[3]:

$$\delta^{13}C_{CH4}\,(T1) = \delta^{13}C_{CH4}\,(T0) - \varepsilon(\ln f);\ [‰] \qquad (6)$$

where T0 and T1 are the beginning and end, respectively, of the incubation and $f$ represents the fraction of $CH_4$ remaining at T1. The predicted $CH_4$ isotope value was −36‰ for both periods June–July and July–August, and is in very good agreement with the values of −36 and −37‰ measured for the incubations at 8 m depth.

**Porewater and sediment sampling.** Sediment cores were taken at Stations A (45 m depth) and D (2.5 m depth) on 11 June 2016 and at S1, S2, S3, S4 (maximum water depth at the stations: 3, 7, 23, and 45 m, respectively) on 29 September 2016 (Supplementary Fig. 1). Sampling was performed with a gravity sediment corer (Uwitech, Mondsee, Austria) equipped with an acrylic liner of 100 cm in length and with an internal diameter of 6 cm. The liner had pre-drilled holes to fit either Rhizons (Rhizosphere Research Products, http://rhizosphere.com/rhizons) or 3 mL syringes at 1 cm intervals, covered with tape.

**Porewater $CH_4$ and $CO_2$ concentrations, and $\delta^{13}C$.** About 3 mL of sediment was sub-sampled at 1–2 cm depth intervals with headless 3-mL syringes through the pre-drilled holes from the selected depths. The sediment sub-sample was immediately placed into 1 L glass bottle (Duran GmbH, Mainz, Germany) containing

600 mL of lake water previously bubbled to reach equilibration with atmospheric air. The subsequent procedure followed the same as for the water column headspace method. Porewater $CH_4$ concentrations were back calculated from the headspace concentrations accounting for dilution of sediments in lake water (assuming that aerated lake water contained 1.75 ppm of $CH_4$ and 410 ppm of $CO_2$), temperature, headspace ratio, and assuming a porosity of 0.9.

**Porewater dissolved organic carbon and $\delta^{13}C$.** Porewater was extracted using Rhizon technology, a hydrophilic, porous polymer tube with 2.5 mm in diameter, 50 mm in length, and 0.12–0.18 µm pore size membrane. Rhizons were inserted into the sediment at a depth resolution of 1 cm. The sample was immediately stored in 2-mL pre-evacuated vials with no headspace. DOC concentration and its stable carbon isotope composition ($\delta^{13}C_{DOC}$ in ‰ vs. VPDB standard) were measured at the UNIL-IDYST by elemental analysis/isotope ratio mass spectrometry (EA/IRMS) using a Carlo Erba 1108 elemental analyzer (Fisons Instruments, Milan, Italy) coupled by a ConFlo III continuous flow open split interface to a Delta V Plus isotope ratio mass spectrometer, both of Thermo Fisher Scientific, Bremen, Germany. Aliquots (1.7 mL) of the porewater samples were freeze-dried in a combusted (480 °C, >4 h) 2 mL glass vial containing 1 mg of combusted (600 °C, >4 h) quartz wool and placed in a tin capsule for EA/IRMS analysis. The precision of the $\delta^{13}C_{DOC}$ measurements by EA/IRMS was better than 0.1‰.

**Sediment total organic carbon content and $\delta^{13}C$.** About 3 mL of sediment was sub-sampled at 1-cm depth intervals with headless 3-mL syringes through the pre-drilled holes. Total organic carbon content and its stable carbon isotope composition ($\delta^{13}C_{TOC}$ in ‰ vs. VPDB standard) were measured from freeze-dried and homogenized sediment samples. The homogenized sediment was pretreated with 12.5% HCl to remove carbonates and TOC was measured using the EA/IRMS system. The DOC and TOC contents were determined from the peak areas of the major isotopes using the calibration standards for $\delta^{13}C$. The precision of the $\delta^{13}C_{TOC}$ measurements by EA/IRMS was better than 0.05‰.

**Methane benthic fluxes.** Methane fluxes at the sediment–water interface were calculated (Stations S1, S2, S3, S4) with Fick's 1st law over the linear top 5 cm decrease of the porewater concentration profile, according to Berner et al.[62]

$$F_{sed} = -\phi \left( D_{CH4}\theta^{-2} \right) \frac{\partial C}{\partial z}; \; [\text{mmol m}^{-2}\text{d}^{-1}] \qquad (7)$$

where $F_{sed}$ is the diffusive $CH_4$ flux at the sediment–water interface, $\phi$ the porosity of the sediments (assumed as 0.9), $D_{CH4}$ the diffusion coefficient for $CH_4$ in water ($1.5 \times 10^{-5}$ cm$^2$ s$^{-1}$)[63], $\theta^2$ the square of tortuosity (1.2)[64] and $\partial C/\partial z$ the measured vertical concentration gradient within the first 5 cm of surface sediments. Negative fluxes indicate a $CH_4$ loss from the sediment.

**Apparent isotopic fractionation of methanogenesis.** The apparent fractionation factor ($\alpha_{app}$) during methanogenesis was defined as in Conrad[31]:

$$\alpha_{app} = \frac{\delta CO_2 + 10^3}{\delta CH_{4source} + 10^3}; \; [-] \qquad (8)$$

where the isotopic signature of source $CH_4$ was estimated by correcting the $\delta^{13}C_{CH4}$ ambient measurement for the isotopic fractionation due to diffusion and oxidation, $\Delta$, as in Bogard et al.[9]:

$$\Delta = (1 - \alpha) \times 10^3 \qquad (9)$$

$\alpha$ was taken from literature as 0.9992 for evasion[65] and 0.98 for MOx[61]. In Table 3, the methanogenic precursor is considered as the precursor of acetate, i.e., organic carbon. This derivation is possible assuming negligible isotopic fractionation during acetate formation[31].

**Mass balance.** The mass balance for the SML (zone 2) was calculated as in Eq. 3, where the water volume ($\forall$) is ~0.04 km$^3$, the sediment surface area ($A_s$) and the planar area ($A_p$) equal ~0.1 km$^2$ and ~8.4 km$^2$, respectively. The individual fluxes: surface flux ($F_S$, Fig. 1), littoral flux ($F_L$, Fig. 1), diffusive vertical flux ($F_Z$, $F_D$, Fig. 1), and riverine input ($F_R = Q_R \times C_R$, Fig. 1) were estimated as described below.

**Surface methane flux.** $CH_4$ flux at the water–air interface ($F_s$) was measured with a floating chamber attached to a portable GHG analyzer (UGGA; Los Gatos Research, Inc.). Instrument-specific precision at ambient concentrations (1–$\sigma$ of 100 s average) for [$^{12}CO_2$] is 40 ppb and for [$^{12}CH_4$] is 0.25 ppb. The floating chamber consisted of an inverted plastic container with foam elements for floatation (as in McGinnis et al.[18]). To minimize artificial turbulence effects, the buoyancy element was adjusted that only ~2 cm of the chamber penetrated below the water level. The chamber was painted white to minimize heating. Two gas ports (inflow and outflow) were installed at the top of the chamber via two 5-m gas-tight tubes (Tygon 2375) and connected to the GHG analyzer measuring the gaseous $CH_4$ concentrations in the chamber every 1 s. Transects were performed with the chamber deployed from the boat (with engine shut down). The boat and chamber

were allowed to freely drift to minimize artificial disturbance. Fluxes were obtained by the slopes of the resolved $CH_4$ curves over the first ~10 min, when the slopes were approximately linear ($R^2 > 0.97$).

The chamber-based $CH_4$ flux measurements were then compared to fluxes estimated based on wind speed. Wind speed-based fluxes were calculated as follows:

$$F_S = k_{CH4} \times kh \times (pCH_{4\,wtr} - pCH_{4\,atm}); \; [\text{mmol m}^{-2}\text{d}^{-1}] \qquad (10)$$

where $pCH_{4\,wtr}$ is the $CH_4$ partial pressures measured in water, $pCH_{4\,atm}$ is the assumed partial pressures of atmospheric $CH_4$ (1.75 ppm), kh is the Henry constant of $CH_4$ dissolution at in situ temperature, and $k_{CH4}$ is the gas transfer velocity. To compute $k_{CH4}$ values, we first derived $k_{600}$ estimates using a wind speed-based relationship. Wind speed was measured at 10 m height ($U_{10}$; m s$^{-1}$ at the nearby Mosen Meteo station, Meteo Group Schweiz AG). We then converted $k_{600}$ to $k_{CH4}$ using:

$$k_{CH4} = k_{600} \left( \frac{Sc_{CH4}}{600} \right)^c; \; [\text{m s}^{-1}] \qquad (11)$$

where $Sc_{CH4}$ is the dimensionless Schmidt number for $CH_4$ (as in Engle and Maleck[66]), $c$ is a wind speed-dependent conversion factor, for which we used −2/3 for $U_{10} < 3.7$ m s$^{-1}$, and −1/2 for all other wind speeds[67]. We further calculate fluxes based on relationships as in MacIntyre et al.[27] for low turbulent regimes. Average flux (April–August 2016) is equal to $0.8 \pm 0.2$ mmol m$^{-2}$ d$^{-1}$ from McIntyre relationship for positive buoyancy and to $0.6 \pm 0.3$ mmol m$^{-2}$ d$^{-1}$ from chamber measurements. The latter, not significantly different from the wind-based relationship, was used for the mass balance (Table 2).

**Littoral methane flux.** Diffusion from the sediment to the water column ($F_L$) was estimated at shallow sites characterized by reed vegetation (Stations S1 and S2 Fig. 1, surface water $CH_4$ ~1 µmol L$^{-1}$) from the $CH_4$ porewater profiles as described above (equal to 1.6 and 1.9 mmol m$^{-2}$ d$^{-1}$, respectively). The contribution to littoral $CH_4$ was estimated from ebullition rates determined at the same sites by Flury et al.[30] assuming bubbles entirely composed by $CH_4$ and that all of the $CH_4$ bubble dissolve into the water ($1.2 \pm 0.8$ mmol m$^{-2}$ d$^{-1}$). For mass balance purposes, the total littoral $CH_4$ flux to the water column was conservatively assumed to be emitted from the whole-lake sediments surface between 0 and 10 m depth.

**Vertical diffusive methane flux in the metalimnion.** The vertical $CH_4$ fluxes $F_z$ (mmol m$^{-2}$ d$^{-1}$) were obtained from $K_z$ (m$^2$ s$^{-1}$) and the $CH_4$ (mmol m$^{-1}$) concentration gradients by Fick's 1st law of diffusion (Eq. 1). Vertical diffusivity $K_z$ was determined in the stratified water below 6 m depth by the heat budget method[26] using temperature measurements from thermistor strings at Station M1 (0, 5, 7.5, 9, 11.5, 14, 17, 20, 25, 35, 46 m) and M2 (0, 5, 7.5, 9, 11.5, 14, 17, 20, 25 m—Supplementary Fig. 2). The temperature mooring was installed from 25 May 2016 to 30 September 2016. The loggers (RBR TR1060, Ottawa, Canada) at 5, 9, and 11.5 m measured temperature every 5 s with a 0.1 s response time and $5 \times 10^{-5}$ °C resolution. The remaining loggers (Vemco Minilog-II-T loggers, Canada) were recording every 1 min, with a resolution of $1 \times 10^{-2}$ °C and response time of <5 min.

**Vertical diffusive $CH_4$ flux from the hypolimnion.** Methane transported by the aeration system from bottom water into the metalimnion ($F_D$) was estimated after McGinnis et al.[18] The used model describes gas transfer across the surface of an individual rising bubble and tracks the dissolution and stripping of $CH_4$. According to the model, small bubbles (4 mm) released from 45 m would have lost all of their $CH_4$ before reaching the thermocline and would thus not contribute $CH_4$ to the metalimnetic peak. However, the aeration system may transport methane from the bottom boundary layer via released air/$O_2$ bubbles. This was estimated as an upper end by implementing the model with the flow rate of the aeration system (180 Nm$^3$ h$^{-1}$) assuming a bubble diameter of 4 mm and a bottom $CH_4$ concentration of 7 µmol L$^{-1}$.

**$CH_4$ input from rivers.** The input of $CH_4$ from rivers ($F_R$) was estimated by the product of the flow rate ($Q_R = 2.5$ m$^3$ s$^{-1}$ for rivers and 1 m$^3$ s$^{-1}$ as conservative average for periodical surface runoff) and the maximum $CH_4$ measured in front of the Aabach river mouth, 1 µmol L$^{-1}$, corrected for background surface lake $CH_4$ ($C_R$) of 0.3 µmol L$^{-1}$.

**Depth-dependent $CH_4$ production rates.** Depth-dependent $CH_4$ changes for the periods 15 June–6 July and 6 July–2 August were calculated by Fick's 2nd law solved for the $CH_4$ production ($P_{gross,m}$) in Eq. 2, where $K_z$ (m$^2$ d$^{-1}$) is the calculated diffusivity. $\frac{\partial C}{\partial t}$ was determined as [$CH_4$ (Time 2)—$CH_4$ (Time 1)] at each depth measured at Station A (mmol m$^{-3}$ d$^{-1}$) and $\frac{\partial^2 C}{\partial z^2}$ is obtained calculating the second derivative of the mean $CH_4$ profile (Time 1, Time 2) at Station A (mmol m$^{-3}$ d$^{-1}$). The net production $P_{net,m}$ (mmol m$^{-3}$ d$^{-1}$) was calculated as $P_{gross,m}-(F_L + F_D)$ with 1 m resolution from 7 to 15 m depth. The top 6 m are

excluded as the air–water interface exchanges are dominated by advection and not diffusion.

**Uncertainty assessment**. The uncertainties of mass balance estimates were assessed by Monte Carlo simulations (999 iterations) (Supplementary Table 1). Each component of the mass balance calculation was randomly picked from either a normal distribution described by the mean and 1 standard deviation values, or within a range. For $F_S$, $F_L$, $F_Z$, one standard deviation (1 SD) on $n$ measurements were assessed while for $P_{gross,m}$ (Eq. 2) the uncertainty was taken as the coefficient of variation of the $CH_4$ concentration profile measurements (as explained in $CH_4$ concentration section). $F_R$ and $F_D$ uncertainties were instead determined as the potential range from 0 to an upper end value (to ensure largely conservative mass balance results) as described in the respective sections.

**Data availability**. All relevant data included in this manuscript are available by request from the authors.

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

## Acknowledgements

We would like to thank Timon Langenegger and Nicole Gallina for their help during the fieldwork and the Canton of Argovia, Department of Civil Engineering, Transportation and Environment for providing access to Lake Hallwil monitoring program database, infrastructure, and boat. Funding for this study was provided by Swiss National Science Foundation (SNSF) Grants No. 200021_160018 (Bubble Flux) and No. 200021_169899 (Methane Paradox).

## Author contributions

D.D. and D.F.M. initiated and designed the study, organized campaigns, performed sampling, and analyzed the data with significant contribution from S.F. and D.V. A.S. was responsible for the Canton Argovia support in data collection, analysis, and logistics. J.E.S. provided porewater and sediment organic carbon stable isotope data analysis and interpretation. D.D. wrote the manuscript with editorial help and conceptual contribution from all the coauthors.

## Additional information

**Competing interests:** The authors declare no competing financial interests.

