## [Peer Review File · Nature Communications]

Reviewers' Comments:

Reviewer #1:

Remarks to the Author:

In this paper, Donis and colleagues provide a new and exciting perspective on a topic of widespread interest among biogeochemists, ecologists and microbiologists. Lakes are an important source of atmospheric methane (CH₄) at the global scale, yet the sources, sinks, and mechanisms driving lake CH₄ emissions are still poorly defined. One of the largest gaps in our understanding of the aquatic CH₄ cycle is to what extent CH₄ emitted to the atmosphere is derived from oxygenated, open water zones. Work that has explored the topic to date has mostly relied on indirect information (correlations of ambient CH₄ to general lake characteristics), or has probed community-scale patterns and the biochemical pathways by which this CH₄ may be generated. Few studies have directly estimated the rates of oxic water CH₄ production or its ecosystem-level relevance. By combining whole-lake mass balancing with detailed spatiotemporal profiling, incubations, and stable isotope methodology in the first study of its kind, the authors identify that a surface layer source of CH₄ is required to account for ~80% of CH₄ emissions. This conclusion is unprecedented in the literature. They identify that metalimnetic CH₄ peaks, generally thought to be the main source of oxic water-derived CH₄, actually contribute little to emissions in this lake, and that because surface-layer CH₄ production largely bypasses microbial oxidation, this pool of CH₄ plays a disproportionate role in supporting emissions. This study provides critical and timely information on this topic that will undoubtedly stimulate widespread interest and further research, as such, Nature Communications seems like an ideal outlet for the work.

The study is very thorough and I do not have major comments regarding study design or field/lab based methodology. My general comments relate to paper presentation, some questions about the mass balancing, and a few questions about the discussion topics. I think these issues are all pretty easily addressed. I have made a bunch of small line by line comments that are all minor, but should help improve the clarity of the paper.

General Comments:

1-Organization of the paper (challenging, given the journal format): I'd cite figure 1 at the end of the introduction where you introduce the approach, so that this very nice figure appears front and center in the paper. It is not really a result so would fit better here I think. I'd also make more explicit reference to it throughout the results and methods to help readers follow the text. The mass balance methods are necessarily very detailed/dense, and the visual reference is a helpful but underused tool.

2-Presentation of the mass balance needs clarification: I had a hard time following the explanation of the mass balance because I kept confusing the 'R' term with the 'P' term. (Lines 172-187, Fig. 6 and 7, and Table 1). If I interpret it correctly, R is the apparent gross production, while P is the net production (can be positive or negative) of CH₄? If yes, then I would change the terms to P_{gross-m}, P_{net-m}, P_{gross-s}, P_{net-s} to refer to gross and net CH₄ production in the metalimnion and in the surface layer, respectively, with all but the 'P' as subscript. This would go a long way to helping readers follow the explanation of the mass balance terms. It would also be nice if these terms could be worked into figure 1 somehow. Maybe this is what the 'ProdCH₄' was meant to be in Fig. 1, but I did not really make that connection. Lastly, where the authors refer to 'apparent production' and 'OMP' throughout the paper, it would be nice to make reference to which term (eg, P_{gross-m}, etc) they are actually talking about. I worry readers will not understand these terms as written.

The equations 3 and 10 describe the mass balances in the metalimnion and surface layer. Why was one in the results and the other in the methods section? I found it hard to keep everything straight and think it would be nice to see both in the same place. In most journals the methods section is the best place to present it, but since the methods are at the end, it may make sense to

show the equations up front, possibly in the legends of tables 1 and 2 and again in the methods?

3-The discussion of mass balance error could be expanded: The authors have done a very nice job addressing some major potential errors (e.g., surface CH₄ contributions from the aeration system, spatial heterogeneity of vertical profiles in the lake, bubble-mediated surface inputs). There are a few additional issues they could consider here: Is there much potential error related to their air-water gas flux estimates? They're such a big term in the surface mass balance that it is worth expanding on, and using these reasonable ranges to further constrain the ~80% contribution value. The chambers and the modelled fluxes diverge considerably on some occasions in Fig. S3. Does this divergence impact the mass balance estimates at all?

One term that could also be evaluated more clearly is the role of oxidation in the surface layer. What numbers were used in the surface layer calculations? Are they in the "P" term in eq. 10? Maybe explore a range of values to provide some context? I am curious as to why the calculated fluxes are so high, but the bottle incubations used for MOX show no big change in CH₄ content in the surface layer? At the end of the day most of these errors likely won't change the story at all, but should be considered.

4- Reworking the section on isotopes in the discussion: This section was a bit confusing. From lines 274-303, the section explaining isotopic signatures of CH₄, OC, and possibly CO₂ was a bit surprising, since table 3 was not introduced in the results. Was there a reason the authors put it here or could it be moved to the end of the results section? It is a nice piece of the paper, but this discussion section could be reworked to present/interpret the findings more coherently. I didn't see any ¹³C-CO₂ data, can this be added to table 3? Second, can the authors back out the actual source ¹³C-CH₄ signature in the surface layer and compare it to that measured in the sediments? They allude to this difference in the paragraph on line 268, but it would be nice to expand on here. The authors could use their known rates (or absence) of MOx, the measured surface ¹³C-CH₄ values, and the fractionation of ~20 per mil, to estimate source ¹³C-Ch₄ in surface. Is the discussion of 'total depletion', on line 280 in the right place? The paragraphs on line 274 and 293 discuss the alpha-app in sediment and the water column. I think these are a slightly different issue than total depletion and could be combined with the paragraph on total depletion presented after.

5- Discussion around eutrophic lakes: The authors devote a few sections of the paper to the potential unimportance of this phenomenon in eutrophic lakes. I had a hard time following the explanation at the end of the discussion (L352-360), and feel this section could be removed or scaled back. It is so speculative that I think it detracts from the important conclusions that the authors can draw here with more certainty.

Specific comments:

L15-18. I'd start with a stronger couple sentences. Why do we care about lake CH₄ emissions?

L18. Can you reword "system analysis"? Whole lake mass balance? Or something like that.

L21. Term 'apparent production' is used throughout but I found it unclear. Can you call it something else more specific?

L22 Isotopic fractionation

L22-23 Can you briefly interpret this result here more deeply? The fractionation effect is different between lake habitats, what does it mean for the source pool of CH₄?

L26-27 Can you take this conclusion further? What are the implications related to identifying this surface CH₄ source?

L32 change 'in' to 'under'

L34 change 'lakes hypolimnia' to 'the hypolimnia of lakes'

L38 may be worth mentioning here that these metalimnetic maxima are thought to be the sites of most intense 'oxic water production'.

L39 why just the metalimnion and not the surface?

L39 – 52 The pgphs seem like they should be combined into 1.

L49 The last sentence off topic, as pgph seems to be about sources of CH₄. I'd work it into the next pgph on the vertical importance and controls of oxidation in lakes.

L54 'Dampening' could be changed to 'restricting' or 'limiting'

L57 In this last sentence, do you mean that this region of the lake, and thus potentially surface water CH₄ production, is more temperature sensitive than CH₄ production at other sites? The reasoning is not totally clear here. Maybe worth referring to the temperature sensitivity of other shallow-water CH₄ emissions pathways to support your claim (i.e., Delsontro et al. 2016 L&O is a good one).

L59 I like this paragraph because it lays out that the study is clearly different from previous efforts. BUT, I would clarify here exactly what they mean by 'oxic' and OMP to ensure readers know what they mean.

L60. Change 'at' to 'under'.

L64. I'd refer to figure 1 after the word 'approach'. See general comment above.

L64. Add 'and isotopic evaluations' after the word 'experiments'.

L67-68. As mentioned in general comments, I think the conclusion ruling out eutrophic lakes could be avoided. I'd instead conclude here by pointing out what this production means for our understanding of the aquatic CH₄ cycle.

L80 change 'since' to 'in'

L112 change 'determined for' to 'measured in'

L114 change 'higher' to 'isotopically enriched'

L116 Why is the surface MO_x rate at 2m depth not presented in fig. 4? I'd add it. Is it just the surface MO_x incubation where the d¹³C signature of CH₄ was unchanged? I'd clarify here how the isotopic composition changed through time in the incubations at each depth. This is very useful information.

L137-139 There were a couple days (Aug then May) in Fig. S3 where the chamber and modelled fluxes deviated. Maybe worth mentioning this here and explaining. Were the days windy in August compared to modelled fluxes that incorporated night-time conditions? What about the following spring when the chambers were low? This variability is relevant in the mass balance so should be explored a bit.

L145 It would be nice to see equation 10 either in this section when you introduce the results for the Zone 2 mass balance, or described in the legend of table 1 so that we can easily extract the meaning of all the numbers in the table, and line them up with the text in part II. As written I found it confusing.

L155 What do you mean by 'strictly diffusive'? Bubble flux is accounted for in another term?

L157 Keep units in nmol to best compare with other reported results

L169 Refer to fig 1 again here and the mass balance eq 10 to ensure readers follow you.

L185 I am likely missing something but CH₄ production (P) does not seem to equal 5nmol/L at 6-7m depth in fig. 6. Can you clarify this? Do we add the bars in Fig. 6, or how do we interpret the figure to follow you to this rate?

L186 Can you explain 1 step further? Why is there net consumption when integrated?

L191 refer to Fig. 4 when presenting this rate?

L192 Is this assumption reasonable? Can you refer to methods or a paper to remind us here?

L201 Refer to Fig. S3 for flux and Fig. 2 for concentration values?

L203 after 'time period' refer to a figure? June and August of when? 2016?

L213 Is the steady state assumption reasonable? Does a change in surface CH₄ through time have any actual implication for your calculations? I doubt it but worth confirming.

L229-236 Table and Fig. references?

L233 Should mention the vertical MO_x patterns here too as an important mechanism.

L237-243. Careful here – ref 9 doesn't fit this paragraph, since cross-treatment differences in GPP were chemically induced, not driven by physical stratification. Ref 9 suggests that there is still a link b/w the oxic water CH₄ production and algal-derived organic matter substrate availability that is in that case independent from physics. In natural environments it is likely that water column stability, algal growth and substrate production, plus CH₄ production are all closely linked. Your DOC-CH₄ correlations may support this notion. This issue is worth explaining here.

L248 The term 'apparent production' could be improved. Maybe say something more explicit, like these vertical peaks of CH₄ in the metalimnion do not actually supply much of the surface-layer CH₄.

L249. Excellent point

L251. Is the OMP here P or R? I have a hard time here – see my general comment regarding the terms.

L252 does production decrease with depth in an integrated way, or between the 2 zones?

L255 Can you expand on this significance? The fact that 3 vastly different approaches agree so well is an incredible result that deserves more attention here.

L263-264 Can you elaborate on the mechanism of MO_x regulation here in a sentence or two?

L268-273 Could expand on why these cross-system differences in surface δ¹³C-CH₄ exist. Also, what might the difference in signature between surface and porewater CH₄ mean? Different organic matter sources? Different biochemical pathways of production? I think it is worth stating this explicitly in this paragraph.

L274-286 Should be combined into 1 paragraph, it is the same topic.

L294 I have a hard time following where the alpha-app values came from here? Can you mention how this was estimated and refer to the data source (or if not presented, mention this).

L311 Do you have a sense of the DIN:TP ratios to expand on algal P limitation idea? See Bergstrom et al. 2008 Aquat. Biol for further context. Not a major point, but could add a nice perspective here.

L314-315 If the MPn comes from the landscape, shouldn't we expect the surface CH₄ production to then decline through time tracking runoff? Seems more likely that a planktonic source of MPn (see Repeta et al. 2016 Nat. Geo) could sustain surface layer CH₄ through the ice free period, with an increase in summer. At the least, I'd finish the paragraph noting multiple potential sources of MPn instead of focusing on agriculture.

L328-333 This sentence is pretty dense. I'd expand this part to detail your calculations better, and walk the readers through this.

L328-333 Also worth reminding readers that your calculated oxic production does agree well with previous experimental estimates of production, which further supports your conclusions.

L348 Worth comparing the 120 mol d⁻¹ to values in Table 2 – this number is very small compared to surface oxic production rates.

L364 Could cite Winder & Schindler 2004 Glob Chg Biol as evidence supporting this lengthened stability period.

L368 'System analysis' could be changed to 'detailed whole lake mass balancing combined with incubation and isotopic approaches to show...'

L386 change 'shaded' to 'sheltered'.

L415 What do you mean by 'longitudinal transect'?

L463 Was -36 per mil at one depth in particular or across the dataset?

L464 Measured values in the lake?

L506 and L 532 I'd refer to figure 1 throughout these parts to help readers follow

L557-569 unclear in this section what method was ultimately used in the mass balance. It is worth stating explicitly how this term was estimated

L572 What is a localized durable hot-spot? How was it determined?

L579 why is this conservative?

L613 List symbol for this depth dependent production.

Display Items:

Fig. 1 – Define Fr from the river in legend

Fig 4 – legend: no CH₄ oxidation, or just a little based on result in line 117. Why are surface MO_x rates not in the bar plots?

Fig 6 – I'd remove CH₄ concentration profiles, they clutter the graph and take away from the main result. Why is the figure not extended to the surface layer? Can you define P and R more clearly in

the legend?

Fig S2 – the range is unclear in legend, why are the profiles not for a specific date?

Fig S5 – Units on axes?

Table S1 – do you mean R, not R^2 ?

Reviewer #2:

Remarks to the Author:

The methane paradox is the notion that methane is produced in oxic surface waters by demethylating processes when seemingly it should be a strictly anaerobic process. I don't doubt that aerobic methane production occurs. But could it be 80% of the methane produced at this study's lake? Despite an admirable argument built by these authors, I remain unconvinced. First, I think that the authors' raise a high bar for themselves. In line 32 they say that the 'methane paradox is an increasingly controversial topic.' Then on line 65, they state that they have "come to the unprecedented conclusion that most of the CH₄ production takes place in the" fully aerobic "surface mixed layer." This sets the stage for a totally convincing documentation of the process, which while the authors do a fine job of inference, they fail to convince.

1. In their conceptual model of the lakes methane "system" the author set the stage by portraying the metalimnion as a methane maximum overlain by a methane poor epilimnion. Most of the methane production is thought to occur in this surface layer, while the metalimnion is a place where methane kind of gets stuck. High concentrations are not indicative of high rates of production. With this I agree.

2. Evidence that bulk of the methane is produced in the surface mixed layer, the epilimnion, is inferred by mass balance, coupled with direct measurements of methane oxidation. The relatively high rates of surface emission cannot be supported by vertical flux across the metalimnion or from the littoral zone so therefore they must be produced in the oxic surface waters (Table 2).

3. While there is certainly a host of detailed interesting measurements in the paper, a couple of things trouble me, mostly focusing on bubble transport and the entrainment of water by this process. In figure 1, the bubbles in the center of the lake are transported from the "diffuser." One is extremely puzzled by this term and doesn't know what to do with it, until perusing the supplemental material hours later, one discovers that the lake is sparged from below. We are assured, however that the sparging is only to prevent anoxia in the deep lake waters, and that it doesn't upset the stratification of the lake itself.

I was troubled by this sparging for two reasons. If the deep lake has so much oxygen demand that it requires sparging to not go anoxic, wouldn't one expect greater rates of methane production in its sediments, and 2) is it not possible that this sparging would set up localized plumes of water advecting upwards carrying methane into the mixed surface layer without upsetting the overall lakes stratification?

4. Figure 4, continuing the thought above, sets up the idea in the two upper panels that the methane maximum is confined to the metalimnion, centered around 5-7 meters depth, but figure 4 shows significant concentration all the way to the surface, nearly 80% in June and 55% in August of the concentrations of methane at the surface relative to the metalimnion. Could there not be a mechanism for bypassing the stratified layer bringing methane into the surface mixed layer and supporting the high flux from the surface?

5. Overall, I think ebullition from the sediments is under represented in this model. While great detail is given on limnology measurements (391-410, water column measurements (412-441), surface methane flux 541-570) the contribution of ebullition is dismissed in three lines 575-577 and referenced away. I don't see that ebullition from deeper zones is considered, despite the high

oxygen demand in the deeper part of the lake. Ebullition has found to be important in many lakes even those not eutrophic or thermakarst. The bubbling rates used by the authors, Flury et al (in ref list) were made on the shores of this lake in a marsh dominated by Phragmites. Emergent aquatic macrophytes are known to suppress ebullition. In the unvegetated portions of the lake shore, ebullition should be higher.

Wik, M., Crill, P. M., Varner, R. K., & Bastviken, D. (2013). Multiyear measurements of ebullitive methane flux from three subarctic lakes. *Journal of Geophysical Research: Biogeosciences*, 118(3), 1307-1321.

Wik, M., Thornton, B. F., Bastviken, D., MacIntyre, S., Varner, R. K., & Crill, P. M. (2014). Energy input is primary controller of methane bubbling in subarctic lakes. *Geophysical Research Letters*, 41(2), 555-560.

I don't suggest that ebullition is supporting surface fluxes, but ebullition, higher lateral methane transport, and sparging of deeper waters may be bringing methane to the surface mixed layer.

6. Incubations. The authors conducted incubations to constrain methane oxidation rates. The results are very interesting showing higher incubation rates in the deep waters where ^{13}C is enriched. I wondered though, the incubations of surface water showed very little change in methane concentration, and these were interpreted that oxidation rates in the surface mixed layer are nil. However, if methane production is so important in this surface mixed layer, why didn't these incubations show increases in methane concentration over time? Maybe the mechanism doesn't work in a bottle? In that case, if the authors want to support their "unprecedented conclusion" about such a controversial topic, in my opinion a direct measurement of methane production would convince me.

7. Isotopes. The lighter isotopes in the surface and enriched ones in the deeper lake waters are consistent with the oxidation pattern measured by the authors. However the lighter surface mixed layer d^{13}C is also consistent with transport from sediments via ebullition. Production of methane from methylated substrates often leads to enriched methane as the substrates are limiting and used to completion. Eg. Kelley, C. A., Poole, J. A., Tazaz, A. M., Chanton, J. P., & Bebout, B. M. (2012). Substrate limitation for methanogenesis in hypersaline environments. *Astrobiology*, 12(2), 89-97.

Reviewer #3:

Remarks to the Author:

Several recent studies provided strong evidence of methane production in oxygenated freshwaters challenging the long-standing paradigm that microbial methane production occurs only under anoxic conditions and forces us to rethink the ecology and environmental dynamics of this powerful greenhouse gas. Conventional explanations for this paradox include input from nearby anoxic sediments and shorelines and production within microanoxic zones in deeper layers of lakes (metalimnion). However, the study by Donis and co-authors provide unambiguous evidence for methane production within the surface mixed layer (epilimnion) under high oxygen concentrations. Thus methane production in the upper oxic water layers places the methane source closer to the air-water-interface, where convective mixing and microbubble detrainment might lead to a methane efflux higher than that previously assumed. Therefore the reported results of this study are novel, highly interesting and important as they further improve our understanding of aerobic methane formation in surface waters of lakes. I strongly favor the publication of this work in *Nature*

Communications. However, to improve the work I have minor comments listed below which the authors should pay attention to. Abstract, line 19-20 "...we reveal that 80 % of CH₄ emissions to the atmosphere (>80 kg d⁻¹) are due to CH₄ produced within the lake surface mixed layer...". This sentence implies that methane emissions are occurring frequently at similar rates throughout the whole year. I presume that the calculated daily average rate of CH₄ emissions and the total amount (~20 tons yr⁻¹) is only valid for the season from May until October. Please revise sentence in the abstract and clarify this issue throughout other parts of the manuscript (results and discussion). Abstract, line 22 "Fractionation between the CH₄ precursors and CH₄ in the surface mixed layer is distinct from sedimentary methane production (30 ‰ vs. 40 ‰)." Sentence is unclear (isotope terminology!), better write "Stable carbon isotope values indicate that CH₄ in the surface mixed layer is distinct from sedimentary CH₄ production". Page 2, line 47. "...(2) algal metabolites..." This part needs slight modification as the study by Lenhart et al. has shown that the marine algae *Emiliana huxleyi* is able to produce CH₄ per se but also from the thio-methyl group of added methionine. In this context I would like to inform the authors about a just published paper that describes a chemical mechanism for iron-oxo-catalyzed methane formation (Comba, P., Benzing, K., Martin, B., Pokrandt, B. and Keppler, F. ()), Nonheme iron-oxo-catalyzed methane formation from methyl thioethers: Scope, mechanism and relevance for natural systems. Chem. Eur. J.. Accepted Author Manuscript. doi:10.1002/chem.201701986). Page 5, Figure 2, unit given for chl a in legend and figure is not consistent (μmol L⁻¹ or μg L⁻¹ ?) Page 15, Table 3. Replace "Total depletion" by "isotope difference between δ¹³CCH₄ and δ¹³CDOC / δ¹³CDOC". Values should be negative. Also rewrite sentence in the manuscript at line 282-284. Statistics: For almost all Figures, Tables and calculations in the text regarding methane fluxes (e.g. daily and total flux) and mass balances from different compartments no statistical values were provided (e.g. error bars, SDs, uncertainty range, number of replicates, etc.). Furthermore, no information about the analytical uncertainties for the measurement systems described in the method section was provided (e.g. stable isotope measurements). This needs to be substantially improved in the revised manuscript.

Reviewer #1 (Remarks to the Author):

In this paper, Donis and colleagues provide a new and exciting perspective on a topic of widespread interest among biogeochemists, ecologists and microbiologists. Lakes are an important source of atmospheric methane (CH₄) at the global scale, yet the sources, sinks, and mechanisms driving lake CH₄ emissions are still poorly defined. One of the largest gaps in our understanding of the aquatic CH₄ cycle is to what extent CH₄ emitted to the atmosphere is derived from oxygenated, open water zones. Work that has explored the topic to date has mostly relied on indirect information (correlations of ambient CH₄ to general lake characteristics), or has probed community-scale patterns and the biochemical pathways by which this CH₄ may be generated. Few studies have directly estimated the rates of oxic water CH₄ production or its ecosystem-level relevance. By combining whole-lake mass balancing with detailed spatiotemporal profiling, incubations, and stable isotope methodology in the first study of its kind, the authors identify that a surface layer source of CH₄ is required to account for ~80% of CH₄ emissions. This conclusion is unprecedented in the literature. They identify that metalimnetic CH₄ peaks, generally thought to be the main source of oxic water-derived CH₄, actually contribute little to emissions in this lake, and that because surface-layer CH₄ production largely bypasses microbial oxidation, this pool of CH₄ plays a disproportionate role in supporting emissions. This study provides critical and timely information on this topic that will undoubtedly stimulate widespread interest and further research, as such, Nature Communications seems like an ideal outlet for the work.

The study is very thorough and I do not have major comments regarding study design or field/lab based methodology. My general comments relate to paper presentation, some questions about the mass balancing, and a few questions about the discussion topics. I think these issues are all pretty easily addressed. I have made a bunch of small line by line comments that are all minor, but should help improve the clarity of the paper.

General Comments:

1-Organization of the paper (challenging, given the journal format): I'd cite figure 1 at the end of the introduction where you introduce the approach, so that this very nice figure appears front and center in the paper. It is not really a result so would fit better here I think. I'd also make more explicit reference to it throughout the results and methods to help readers follow the text. The mass balance methods are necessarily very detailed/dense, and the visual reference is a helpful but underused tool.

We agree with the Reviewer's suggestion, and have inserted Figure 1 at the end of the introduction. Also the number of references to it was substantially increased throughout the manuscript.

2-Presentation of the mass balance needs clarification: I had a hard time following the explanation of the mass balance because I kept confusing the 'R' term with the 'P' term. (Lines 172-187, Fig. 6 and 7, and Table 1). If I interpret it correctly, R is the apparent gross production, while P is the net production (can be positive or negative) of CH₄? If yes, then I would change the terms to P_{gross-m}, P_{net-m}, P_{gross-s}, P_{net-s} to refer to gross and net CH₄ production in the metalimnion and in the surface layer, respectively, with all but the 'P' as subscript. This would go a long way to helping readers follow the explanation of the mass balance terms. It would also be nice if these terms could be worked into figure 1 somehow. Maybe this is what the 'ProdCH₄' was meant to be in Fig. 1, but I did not really make that connection. Lastly, where the authors refer to 'apparent production' and 'OMP' throughout the paper, it would be nice to make reference to which term (eg, P_{gross-m}, etc) they are actually talking about. I worry readers will not understand these terms as written.

We realize that P, R, OMP terminology may be confusing and are thankful for the suggestions to clarify their expression. We thought of using net/gross subscripts for the production terms P and R in the mass balance calculations but hesitated given that R does not represent exactly a gross production. Indeed R, obtained from 2nd Fick's law, expresses a production rate that includes losses to vertical diffusion (F_z) and methane oxidation (MOx) as well as CH₄ addition due to lateral transport (F_L). However, we will now use the more straight forward net/gross terminology and replace R with P_{gross,m} and P with P_{net,m} and P_{net,s} for zone 1 and 2, respectively.

—These terms are introduced in the following lines:

L146: *"To determine the overall net production in the lake (P_{net} , Fig. 1), mass balances were performed within the two defined zones using the various rates shown in Table 1 and 2."*

L 176-178: *"With Fick's 2nd law, we determined the depth dependent CH₄ production ($P_{z,gross,m}$) in zone 1 expressed by the sum of losses by diffusion (F_z , Fig. 1) and oxidation and inputs from the littoral zone and from the hypolimnion (F_L and F_D ; Fig. 1)."*

L187: *"Local methane production ($P_{net,m}$) was calculated by removing the estimated littoral contributions as $P_{gross,m} - (F_L + F_D)$ (Fig. 1), for both periods June-July and July-August 2016..."*

L208: *"we assume on the seasonal scale that the surface layer can be modeled as a well-mixed reactor, and CH₄ net production rates ($P_{net,s}$) can be estimated as follows..."*

We now use the term P_{net} as well in Figure 1, while keeping the expression OMP when referring to the phenomenon of oxic methane production in other studies (lines: 276, 291, 294, 315, 328, 357, 407).

The equations 3 and 10 describe the mass balances in the metalimnion and surface layer. Why was one in the results and the other in the methods section? I found it hard to keep everything straight and think it would be nice to see both in the same place. In most journals the methods section is the best place to present it, but since the methods are at the end, it may make sense to show the equations up front, possibly in the legends of tables 1 and 2 and again in the methods?

For clarity, the equation used for mass balance in zone 1 is Eq. 2 and the one for zone 2 is now Eq. 3.

These are now both in the results when they are first referred to in lines 176-180, 208-214. As per the reviewer's suggestion, they were therefore removed from the methods section.

Furthermore, as suggested by the reviewer, we now re-list equation 2 and 3 in Table 1 and 2 captions, respectively.

3-The discussion of mass balance error could be expanded: The authors have done a very nice job addressing some major potential errors (e.g., surface CH₄ contributions from the aeration system, spatial heterogeneity of vertical profiles in the lake, bubble-mediated surface inputs). There are a few additional issues they could consider here: Is there much potential error related to their air-water gas flux estimates? They're such a big term in the surface mass balance that it is worth expanding on, and using these reasonable ranges to further constrain the ~80% contribution value.

The fact that CH₄ air water exchange is the most relevant term is absolutely true; we agree to expand on the uncertainties related to its estimate. To assess overall uncertainties in the mass balance(s), we ran Monte Carlo simulations (999 iterations), in which each component in the calculation is randomly picked from a normal distribution around its mean value (F_S , F_L , F_Z , $P_{\text{gross,m}}$), or from a range going from 0 to a maximum estimate (F_R , F_D). L672-680.

The following Table was added to the supplemental material (Supplementary Table 2) and summarizes mass balance(s) components mean and standard deviations (s.d.) as well as the number of samples and the time (time range) when measurements were performed.

Supplementary Table 2

Mass balance component	Mean/range	s.d., N, unit	Measurement
F_s (Evasion from surface)	0.6	0.3 (28) $\text{mmol m}^{-2} \text{d}^{-1}$	April-Aug 2016
$F_L(\text{sed})$ (Diffusion from littoral sediments)	1.75	0.2 (2) $\text{mmol m}^{-2} \text{d}^{-1}$	Sept 2016
$F_L(\text{eb})$ (Dissolution from littoral ebullition)	1.2	0.8 (8) $\text{mmol m}^{-2} \text{d}^{-1}$	Flury et al. 2010
F_z (Diffusion from metalimnion peak)	0.03	0.01 (3) $\text{mmol m}^{-2} \text{d}^{-1}$	June-August 2016
F_D (surface CH ₄ contributions from the aeration system)	0-0.014	$\text{mmol m}^{-2} \text{d}^{-1}$	
F_R (Input from rivers)	0-0.005	$\text{nmol L}^{-1} \text{d}^{-1}$	
$P_{\text{gross-m}}$ (Production from Eq. 2)	0.01	0.001 $\text{nmol L}^{-1} \text{d}^{-1}$	

By this range of uncertainties, we determined a plausible $P_{\text{net-s}}$ deviation from the mean ($110 \pm 60 \text{ nmol L}^{-1} \text{d}^{-1}$), that is largely ascribable to the air-water flux s.d.

Note:

- F_D (surface CH₄ contributions from the aeration system): the upper end of the range describes gas transfer across the surface of an individual rising bubble with the flow rate of the aeration system ($180 \text{ Nm}^3 \text{ h}^{-1}$) assuming a bubble diameter of 4 mm and a bottom CH₄ concentration of $7 \mu\text{mol L}^{-1}$ (maximum measured at 48m).

- F_R (input from rivers): the upper end of the range corresponds to input of CH₄ from rivers, estimated by the product of the flow rate ($Q_R = 2.5 \text{ m}^3 \text{ s}^{-1}$ for rivers and $1 \text{ m}^3 \text{ s}^{-1}$ as conservative average for periodical surface runoff) and the maximum CH₄ measured in front of the Aabach river mouth ($1 \mu\text{mol L}^{-1}$), corrected for background surface lake CH₄ of $0.3 \mu\text{mol L}^{-1}$.

- $P_{\text{gross,m}}$ (gross methane production obtained in the metalimnion between 5-10 meters): standard deviation was set as 10 %. i.e. the coefficient of variation of CH₄ concentration profile measurements.

The chambers and the modelled fluxes diverge considerably on some occasions in Fig. S3. Does this divergence impact the mass balance estimates at all?

We actually believe the surface flux should be higher than what is reported. With one exception, all flux measurements were conducted during the daytime, when the buoyancy flux is greater than zero (i.e. stably stratified). At night, when the atmosphere cools, the buoyancy flux would typically turn negative. This would nearly double the flux estimates at night during convective mixing (MacIntyre 2010). In addition, there could be microbubble transport enhancement which would further enhance the methane flux estimates (McGinnis et al. 2014). In this way, we are very conservative in our methane flux estimates and increases will increase the estimated surface methane production estimates.

We added to the figure the flux estimates during convective mixing, and expanded on the 2 “outlying” chamber measurements. Please note that Supplementary Fig. 3 is now Supplementary Fig. 4.

Furthermore, we have added the following lines to the discussion:

L139-143: “The flux estimate for negative buoyancy, typical for night convective mixing, is nearly double than what was estimated from chamber measurements as these were almost always taken during the day (Supplementary Fig. 4). Consequently, the surface flux component (F_s , Fig. 1) is a conservative estimate for the summer period.”

One term that could also be evaluated more clearly is the role of oxidation in the surface layer. What numbers were used in the surface layer calculations? Are they in the “P” term in eq. 10?

In the previous version of the manuscript, we neglected the methane oxidation term in mass balance for the surface layer, as we have not measured any isotope change after the in situ incubations at 2 and 8 meters (Fig. 4). We now add the oxidation term in the mass balance as a potential MOx, as described in the next comment.

Maybe explore a range of values to provide some context? I am curious as to why the calculated fluxes are so high, but the bottle incubations used for MOX show no big change in CH₄ content in the surface layer? At the end of the day most of these errors likely won't change the story at all, but should be considered.

From in situ incubations in the SML we measured a decrease of 0.3 nmol CH₄ L⁻¹ d⁻¹ (June- July), and 3.6 ± 0.2 nmol CH₄ L⁻¹ d⁻¹ (July-August), but contrary to the same measurements below 10 m, no isotope enrichment was observed. We therefore assumed negligible oxidation rates for surface layer mass balance calculations. This assumption is supported in the literature (e.g. Murase et al., 2005) due to light inhibition. For clarity we add the MOX contribution to losses in Zone 2 (surface mixed layer, 150 ± 8 mol d⁻¹). As predicted by the reviewer it has very little consequences on the budget, accounting for only 3 % of the total net production P_{net,s}.

CH ₄ flux	Description	mol d ⁻¹	kg d ⁻¹	nmol L ⁻¹ d ⁻¹
F _S	Evasion from surface	5,040 ± 2,520	80 ± 40	121 ± 60
MOx	Methane oxidation	150 ± 8	2 ± 0.1	3.6 ± 0.2
F _L (eb)	Dissolution from littoral ebullition	134 ± 89	2 ± 1	3 ± 2
F _L (sed)	Diffusion from littoral sediments	196 ± 22	3 ± 0.3	5 ± 0.5
F _R	Input from rivers	0 - 207	0-3	0-5
F _Z	Diffusion from metalimnion peak	252 ± 84	4 ± 1	6 ± 2
P _{net-s}	Local production	4,600 ± 2,500	73 ± 40	110 ± 60

Additionally, we added the following to the Discussion.

L267-270 : "Bottle incubations in the SML show either negligible CH₄ oxidation (0.3 nmol L⁻¹ d⁻¹, June-July 2016) or higher oxidation (3.6 ± 0.2 nmol L⁻¹ d⁻¹, July-August 2016) with no change in isotope values (2‰ after 1-month incubation). This may indicate oxidation is compensated by a CH₄ production mechanism."

L304-307: "With a δ¹³C_{CH4} equal to -62 ‰ at Time 1, as from in situ measurements, the isotope value associated to the measured consumption rates of 3 nmol L⁻¹ d⁻¹ should have been in the order of -41 ‰ instead of the final measured -64 ‰ (assuming MOx with fractionation factor of 20 ‰)."

4- Reworking the section on isotopes in the discussion: This section was a bit confusing. From lines 274-303, the section explaining isotopic signatures of CH₄, OC, and possibly CO₂ was a bit surprising, since table 3 was not introduced in the results. Was there a reason the authors put it here or could it be

moved to the end of the results section? It is a nice piece of the paper, but this discussion section could be reworked to present/interpret the findings more coherently. I didn't see any ^{13}C - CO_2 data, can this be added to table 3?

We agree the isotope ratio results should go in the Results section (now L229-250), while we also leave in the discussion the more speculative part on the possible precursors for methanogenesis (L314-324). As the $\delta^{13}\text{C}_{\text{CO}_2}$ are used to calculate the apparent fractionation we added the values to Table 3.

Second, can the authors back out the actual source ^{13}C - CH_4 signature in the surface layer and compare it to that measured in the sediments? They allude to this difference in the paragraph on line 268, but it would be nice to expand on here. The authors could use their known rates (or absence) of MOx , the measured surface ^{13}C - CH_4 values, and the fractionation of ~ 20 per mil, to estimate source ^{13}C - CH_4 in surface.

Below is how we performed the calculation, and now detailed it in the results (L239-241). For MOx we used the rates from literature (Bastviken 2002, as in Methods) as our rates are confounded by the lack of change in isotope data after incubations.

L 243-244 and L570-564: *"the isotopic signature of source CH_4 was estimated by correcting the $\delta^{13}\text{C}_{\text{CH}_4}$ ambient measurement for the isotopic fractionation due to diffusion and oxidation, Δ , as in Bogard et al.: $\Delta = (1 - \alpha) \times 10^3$, α was taken from literature as 0.9992 for evasion and 0.98 for MOx . "*

Is the discussion of 'total depletion', on line 280 in the right place? The paragraphs on line 274 and 293 discuss the alpha-app in sediment and the water column. I think these are a slightly different issue than total depletion and could be combined with the paragraph on total depletion presented after.

We agree, and the paragraph was rearranged as (L230-235): *"To assess possible similarity between water column and porewater CH_4 formation we investigate the difference between the isotope measurements of CH_4 and methanogenic precursors (total and dissolved organic carbon, TOC and DOC). Based on calculations according to Bogard et al. ⁹ (see Methods) we infer a smaller difference between the isotope values of carbon source and CH_4 produced in the oxic water column (-32 – -29 ‰) as compared to the sediment methanogenesis (- 44 – - 41 ‰, Table 3)."*

5- Discussion around eutrophic lakes: The authors devote a few sections of the paper to the potential unimportance of this phenomenon in eutrophic lakes. I had a hard time following the explanation at the end of the discussion (L352-360), and feel this section could be removed or scaled back. It is so

speculative that I think it detracts from the important conclusions that the authors can draw here with more certainty.

We agree and removed speculations on eutrophic systems.

Specific comments:

L15-18. I'd start with a stronger couple sentences. Why do we care about lake CH₄ emissions?

We now change to (L16-18) *"Oxic lake surface waters are frequently oversaturated with methane (CH₄). The contribution of this phenomenon to the global CH₄ cycle is significant, thus leading to an increasing number of studies and stimulating debates."*

L18. Can you reword "system analysis"? Whole lake mass balance? Or something like that.

We replaced "system analysis" with "mass balance" in line 18 and elsewhere in throughout the manuscript.

L21. Term 'apparent production' is used throughout but I found it unclear. Can you call it something else more specific? Maybe use "metalimnetic production" or "metalimnetic accumulation"?

We replace "apparent production" with "...physically driven accumulation" in line 21.

L22 Isotopic fractionation. Please see next comment.

L22-23 Can you briefly interpret this result here more deeply? The fractionation effect is different between lake habitats, what does it mean for the source pool of CH₄?

The sentence in line 23-24 reads now as: *"Stable carbon isotope values indicate that CH₄ in the SML is distinct from sedimentary CH₄ production, **suggesting distinct pathways or precursors.**"*

More text would mean deleting other parts as the abstract limit is 150 words.

L26-27 Can you take this conclusion further? What are the implications related to identifying this surface CH₄ source?

As to the above comment we state that this *"suggests distinct pathways or precursors"* (line 24).

L32 change 'in' to 'under'. This was changed (now line 29).

L34 change 'lakes hypolimnia' to 'the hypolimnia of lakes' . Was changed accordingly (now line 31).

L38 may be worth mentioning here that these metalimnetic maxima are thought to be the sites of most intense 'oxic water production'.

We now added to line 32-36 the following: *“However, metalimnetic CH₄ maxima, **thought to be the most intense location for ‘oxic water production’**, were found in a number of oligotrophic to mesotrophic lakes including Lake Stechlin, Germany³, Lake Lugano, Switzerland⁴ and Lake Biwa, Japan⁵, with concentrations several orders of magnitude higher than CH₄ maxima reported for ocean surface oxic waters⁶*

L39 why just the metalimnion and not the surface?

We have changed as “surface oxic waters” in line 36.

L39 – 52 The pgphs seem like they should be combined into 1.

The paragraphs were merged, now lines 37-47.

L49 The last sentence off topic, as pgph seems to be about sources of CH₄. I'd work it into the next pgph on the vertical importance and controls of oxidation in lakes.

We agree and moved the sentence to L 48-52.

L54 'Dampening' could be changed to 'restricting' or 'limiting'

L49: the word "dampening" was substituted with "limiting".

L57 In this last sentence, do you mean that this region of the lake, and thus potentially surface water CH₄ production, is more temperature sensitive than CH₄ production at other sites? The reasoning is not totally clear here. Maybe worth referring to the temperature sensitivity of other shallow-water CH₄ emissions pathways to support your claim (i.e., Delsonetro et al. 2016 L&O is a good one).

The previous sentence *“Therefore, as lakes appear to be rapidly responding to climate warming, it is essential to understand feedbacks and dynamics of this shallow CH₄ source.”* Was meant to address the fact that a stronger water column stratification (as a consequence of warming climate) would enhance the effect of “isolation” between surface mixed layer (where we find most of the methane production) and the metalimnion (where methane oxidation seems more efficient). We therefore reformulate the sentence as (L52-54) as: *“Furthermore, the often observed absence or inhibition of CH₄-oxidizers in the*

epilimnion of lakes^{19,20} is likely to be particularly significant in this context, indirectly acting to sustain high CH₄ concentrations and subsequent emissions.”

L59 I like this paragraph because it lays out that the study is clearly different from previous efforts. BUT, I would clarify here exactly what they mean by ‘oxic’ and OMP to ensure readers know what they mean. The part reads now as (L54-57): *“While there is an increasing number of publications on ‘oxic’ methane production (OMP; in the sense of Tang et al.⁶. i.e. “without inferring whether the biochemical pathway itself requires oxygen”) in lakes, no studies have so far addressed the associated rates under in situ conditions.”*

L60. Change ‘at’ to ‘under’. This was done (L57).

L64. I’d refer to figure 1 after the word ‘approach’. See general comment above.

We add(Fig. 1) to line 61.

L64. Add ‘and isotopic evaluations’ after the word ‘experiments’. This was done (L62).

L67-68. As mentioned in general comments, I think the conclusion ruling out eutrophic lakes could be avoided. I’d instead conclude here by pointing out what this production means for our understanding of the aquatic CH₄ cycle.

We now change the sentence as (L64-66): *“This significant source of CH₄ is in direct contact with the atmosphere, implying an important contribution to global budgets estimates of this potent greenhouse gas.”*

L80 change ‘since’ to ‘in’. This was done (L75).

L112 change ‘determined for’ to ‘measured in’. This was done (L 114).

L114 change ‘higher’ to ‘isotopically enriched’?

L 115: *“Seasonally, the CH₄ in the Lake Hallwil surface layer had an average $\delta^{13}\text{C}_{\text{CH}_4}$ (June-August 2016) of $-60 \text{‰} \pm 2 \text{‰}$ (s.d.), and becomes isotopically enriched ($\sim -40 \text{‰}$) below the CH₄ peak and thermocline (Fig. 4).”*

L116 Why is the surface MOx rate at 2m depth not presented in fig. 4? I'd add it. Is it just the surface MOx incubation where the d13C signature of CH4 was unchanged? I'd clarify here how the isotopic composition changed through time in the incubations at each depth. This is very useful information.

We agree and modified the figure as to include the surface incubation results:

Figure 4: Water column CH₄ concentration, $\delta^{13}\text{C}_{\text{CH}_4}$ values and oxidation rates. (a, c, e) Water column profiles of CH₄, $\delta^{13}\text{C}_{\text{CH}_4}$ and temperature for 17 June, 7 July and 3 August 2016. CH₄ profiles show the distinctive metalimnetic maxima while at 25–30 m concentrations were $<0.05 \mu\text{mol L}^{-1}$. The $\delta^{13}\text{C}_{\text{CH}_4}$ profiles in the epilimnion are rather uniform with values around $-62 - -60 \text{‰}$. (b, d) Average dissolved oxygen profiles and MOx rates obtained from *in situ* incubations for the periods June-July 2016 and July-August 2016 show maximum oxidation rates and correspondingly higher $\delta^{13}\text{C}_{\text{CH}_4}$ values at 15 and 13 m depth, respectively. In the SML, oxidation rates were negligible for the period June- July 2016, and higher during the period July-August 2016 ($3.6 \pm 0.2 \text{ nmol L}^{-1}\text{d}^{-1}$) although associated with no significant change in $\delta^{13}\text{C}_{\text{CH}_4}$ (*).

L137-139 There were a couple days (Aug then May) in Fig. S3 where the chamber and modelled fluxes deviated. Maybe worth mentioning this here and explaining. Were the days windy in August compared to modelled fluxes that incorporated night-time conditions? What about the following spring when the chambers were low? This variability is relevant in the mass balance so should be explored a bit.

As mentioned above, we added to the figure the flux estimates during convective mixing, and expanded on the 2 “outliers” of chamber measurements. Now in Supplementary Fig. 4 caption reads as: “The two chamber measurement series for 12 August 2015 and 15 May 2016 stand out of the wind-based estimates as they were obtained during a particularly windy, cool day and a very warm day with little wind, respectively.”

Below we show (only here) the T profile on May 15, showing a shallow stratification due to a particularly warm day with very low wind (thus very low fluxes to the atmosphere), and the T mooring data from

August 2016 (T loggers at 8,12,20,45 m), where is visible that a strong wind event affected the water column, and thus deepened the mixed layer right before the 12th of August.

L145 It would be nice to see equation 10 either in this section when you introduce the results for the Zone 2 mass balance, or described in the legend of table 1 so that we can easily extract the meaning of all the numbers in the table, and line them up with the text in part II. As written I found it confusing.

We added the equation in Table 2 legend.

L155 What do you mean by 'strictly diffusive'? Bubble flux is accounted for in another term?

We now reformulate the sentence (L158-160) to :

"The vertical transport of dissolved CH₄ from the metalimnion to the SML and hypolimnion is driven via turbulent diffusivity and the concentration gradients, where the basin-scale diffusivity, K_z, was determined to be ..."

L157 Keep units in nmol to best compare with other reported results

Units were changed: Line 166: " ...to be of ~14 nmol L⁻¹ d⁻¹ (0.07 mmol m⁻² d⁻¹) "

L169 Refer to fig 1 again here and the mass balance eq 10 to ensure readers follow you.

We add "Fig. 1" to line 171 and 175.

L185 I am likely missing something but CH₄ production (P) does not seem to equal 5nmol/L at 6-7m depth in fig. 6. Can you clarify this? Do we add the bars in Fig. 6, or how do we interpret the figure to follow you to this rate?

5 nmol L⁻¹ d⁻¹ is the average between red bars in Fig. 6 at 6-7 m for both periods. To be more precise we now changed this to 5 ± 5 nmol L⁻¹ d⁻¹. The paragraph reads now as (186-190): “Local methane production ($P_{net,m}$) was calculated by removing the estimated littoral contributions as $P_{gross,m} - (F_L + F_D)$ (Fig. 1), for both periods June-July and July-August 2016 (Fig. 6), indicating an average (June-August) “oxic” methane production ($P_{net,m}$) of $\sim 5.0 \pm 5.0$ nmol L⁻¹ d⁻¹ between 6 – 7 m. Yet, when net P rates are integrated over the metalimnion (zone 1 in Fig. 6), $P_{net,m}$ become nearly negligible at 0.3 ± 3.0 nmol L⁻¹ d⁻¹ (Table 1).”

L186 Can you explain 1 step further? Why is there net consumption when integrated?

This is from the sum of the production rates (red bars in Figure 6) over 6-10 m for both time periods. The sum is a net consumption of 2 nmol L⁻¹ d⁻¹. Now the text changed to be more precise (since we added the uncertainties) to “nearly negligible rates of 0.3 ± 3 nmol L⁻¹ d⁻¹ (Table 1).” In line 190.

L191 refer to Fig. 4 when presenting this rate?

Figure 4 was added to the rate in line 204.

L192 Is this assumption reasonable? Can you refer to methods or a paper to remind us here?

We agree it needs a reference and add to line 195 (29) DelSontro et al., 2016.

Del Sontro et al., L&O, 2016 – “Methane ebullition and diffusion from northern ponds and lakes regulated by the interaction between temperature and system productivity” - DOI: 10.1002/lno.10335.

Cit: “A strong depth-dependence of ebullition has also been reported elsewhere (Casper et al. 2000; Bastviken et al. 2004; Wik et al. 2013), although the depth at which ebullition ceases varies between ecosystem types and studies. “

We have also confirmed this with the absence of bubbles in our deep retrieved sediment cores, and very low bottom water concentrations. L80-83: “*The aggressive restoration measures were extremely effective, resulting in a re-oligotrophication of the lake where bottom waters remain near completely oxic and preventing methane ebullition from developing in the hypolimnetic sediment (Supplementary Fig. 1).*”

L201 Refer to Fig. S3 for flux and Fig. 2 for concentration values?

We now refer to the figures in line 204-205.

“L203 after ‘time period’ refer to a figure? June and August of when? 2016?

We added to line 207 “ (both 2015-2016).”

L213 Is the steady state assumption reasonable? Does a change in surface CH₄ through time have any actual implication for your calculations? I doubt it but worth confirming.

Over the seasonal time-scale this system analysis, steady state is a reasonable assumption. Surface concentrations change very little ($0.3 \pm 0.1 \mu\text{mol L}^{-1}$). Drops in concentrations would result from sustained high fluxes (high wind, cooling etc.), likewise, higher concentrations would build up resulting from lower winds. These are linear processes (Fick’s Law, and k assuming MacIntyre relations) and are therefore well-reflected in the average. Furthermore, the sensitivity is now included in the Monte Carlo analysis.

We now add for precision to L217: “ ***As surface concentrations do not vary much seasonally (Fig. 2 and Fig. 4), we assume steady state ($\frac{\partial C}{\partial t} \forall = 0$) and solve the mass balance in the top 5 m.***”

L229-236 Table and Fig. references?

References were added to -now- L259-264: “ *The observed metalimnetic CH₄ production rate ($P_{gross,m}$) of about $10 \text{ nmol L}^{-1} \text{ d}^{-1}$ can be largely explained by lateral transport from the adjacent sediments (Table 1). Negative production rates below 10 m (Fig. 6) are explained by oxidation of CH₄ as confirmed by bottle incubations ($\sim 6 \text{ nmol L}^{-1} \text{ d}^{-1}$) where a CH₄ stable carbon isotope ratio increase of 20 ‰ was observed after 1 month for both periods June-July and July-August (Fig. 4).*”

L233 Should mention the vertical MOx patterns here too as an important mechanism.

Accordingly, we added to line 267-270: “*Bottle incubations in the SML show either negligible CH₄ oxidation ($0.3 \text{ nmol L}^{-1} \text{ d}^{-1}$, June-July 2016) or higher oxidation ($3.6 \pm 0.2 \text{ nmol L}^{-1} \text{ d}^{-1}$, July-August 2016) with no change in isotope values (2‰ after 1-month incubation). This may indicate oxidation is compensated by a CH₄ production mechanism.*”

L237-243. Careful here – ref 9 doesn’t fit this paragraph, since cross-treatment differences in GPP were chemically induced, not driven by physical stratification. Ref 9 suggests that there is still a link b/w the oxic water CH₄ production and algal-derived organic matter substrate availability that is in that case independent from physics. In natural environments it is likely that water column stability, algal growth

and substrate production, plus CH₄ production are all closely linked. Your DOC-CH₄ correlations may support this notion. This issue is worth explaining here.

We agree and we removed reference n. 9 from this sentence while adding it to the DOC part in line 353-356: *"Lake Hallwil is surrounded by an intense agricultural landscape which is potentially a source for MPn, although the absence of a strong lateral CH₄ gradient points towards the relationship with DOC (Supplementary Fig. 5b) supporting a link between the oxic water CH₄ production and algal-derived organic matter substrate availability⁹."*

L248 Maybe say something more explicit, like these vertical peaks of CH₄ in the metalimnion do not actually supply much of the surface-layer CH₄.

We now use the proposed definition and state in line 283-284: *"This supports that the highest production rates are expressed at the ventilated surface layer while the CH₄ in the metalimnion represents only a local accumulation that supplies very little CH₄ to the surface-layer."*

L249. Excellent point. Thank you

L251. Is the OMP here P or R? I have a hard time here – see my general comment regarding the terms.

We replaced P with P_{net,m} and R with P_{gross,m}. Please note that P gross exists only for the mass balance in zone 1 because for zone 2 we determine directly the P_{net} from eq. 3.

L252 does production decrease with depth in an integrated way, or between the 2 zones?

This is a very good point. As from Eq.3 and Fig.6, the production over surface is "smeared" over the 5-m surface mixed layer. Production below is much lower as sketched in Fig 1 (that we now add to Line 288). We don't think that the resolution of this analysis supports inferring more information other than that they vary between the layers substantially (100 times).

L255 Can you expand on this significance? The fact that 3 vastly different approaches agree so well is an incredible result that deserves more attention here.

As the reviewer already states these are 3 approaches that are substantially different. Therefore, we expanded with the addition of a sentence that gives the right perspective to such comparison:

(L292-295) *"Despite the different approaches, methane production rates lay within a surprisingly narrow range. Thus our results both support the growing body of evidence for OMP as well as better constrains the rates now reported in multiple freshwater environments."*

L263-264 Can you elaborate on the mechanism of MOx regulation here in a sentence or two?

We now add to line starting 302-306: *“Methane oxidizing bacteria favor isotopically lighter CH₄, leaving a residual CH₄ with a higher δ¹³C_{CH4} value. Both high oxygen concentrations and light exposure have been shown to significantly inhibit MOx^{19,35,36}. In Lake Hallwil, these findings are supported by negligible oxidation rates measured in situ (Jun- July 2016) and by the lighter δ¹³C_{CH4} values above 6 m (Fig. 7).”*

L268-273 Could expand on why these cross-system differences in surface d13C-CH4 exist. Also, what might the difference in signature between surface and porewater CH4 mean? Different organic matter sources? Different biochemical pathways of production? I think it is worth stating this explicitly in this paragraph.

We state in line 322: *“Yet the carbon isotope composition of CH₄ can be influenced by the type of acetate precursor⁴², the production mechanism(s) and pathways...”*

And few lines below (line 330-336):

*“Our estimates for the water column SML CH₄ production show a smaller fractionation factor compared to the sediments (α_{app} = 1.045), but still characteristic for acetoclastic methanogenesis and similar to what was found by Bogard et al.⁹ for their enclosure experiments (α_{app} = 1.02 – 1.04). Furthermore, we assessed a smaller difference between isotope values of precursors (...) **which may indicate that water column P_{net} derives from a distinct pathway, not linked to the sediments.**”*

L274-286 Should be combined into 1 paragraph, it is the same topic. This was done (L281-287).

L294 I have a hard time following where the alpha-app values came from here? Can you mention how this was estimated and refer to the data source (or if not presented, mention this).

This is now better explained in the results (L242-250): “

“The apparent fractionation factor (α_{app}) during methanogenesis was defined as in Conrad et al.³² where the isotopic signature of source CH₄ was estimated by correcting the δ¹³C_{CH4} ambient measurement for the isotopic fractionation due to diffusion and oxidation (see methods). → L 566

Apparent isotopic fractionation of methanogenesis. The apparent fractionation factor (α_{app}) during methanogenesis was defined as in Conrad(32): $\alpha_{app} = \frac{\delta^{13}C_{CO_2} + 10^3}{\delta^{13}C_{CH_4source} + 10^3}$, where the isotopic signature of source CH₄ was estimated by correcting the δ¹³C_{CH4} ambient measurement for the isotopic fractionation due to diffusion and oxidation, Δ, as in Bogard et al.(2014): $\Delta = (1 - \alpha) \times 10^3$. α was taken from literature as 0.9992 for evasion and 0.98 for MOx. “

L311 Do you have a sense of the DIN:TP ratios to expand on algal P limitation idea? See Bergstrom et al. 2008 Aquat. Biol for further context. Not a major point, but could add a nice perspective here.

DIN:P ratio in Hallwil in August 2015 was >100. We now added the information in the following paragraph (see next comment).

L314-315 If the MPn comes from the landscape, shouldn't we expect the surface CH₄ production to then decline through time tracking runoff? Seems more likely that a planktonic source of MPn (see Repeta et al. 2016 Nat. Geo) could sustain surface layer CH₄ through the ice free period, with an increase in summer. At the least, I'd finish the paragraph noting multiple potential sources of MPn instead of focusing on agriculture.

We are thankful for this comment and reshaped lines 345-356 that read as: *"MPn can derive from anthropogenic activity (e.g. herbicide glyphosate) and is known to contribute to the phosphonate pool in lakes and their watersheds⁴⁷. It was furthermore shown that, when environmentally limited, phosphate can be regenerated from semi-labile dissolved organic matter (DOM) through the C-P lyase pathway with formation of CH₄⁴⁸. Indeed, the expression of the C-P lyase gene found in many freshwater cyanobacteria^{49,50} is induced by P limitation⁵¹. This hypothesis suits Lake Hallwil mainly for two reasons: 1) low water column P concentrations (3 μg L⁻¹ in top 5 m, DIN:P_{tot} > 100 in August 2015) and 2) surface (1 m) CH₄ concentrations correlate with dissolved organic carbon (DOC) (Supplementary Fig. 5a). Lake Hallwil is surrounded by an intense agricultural landscape which is potentially a source for MPn, although the absence of a strong lateral CH₄ gradient points towards the relationship with DOC (Supplementary Fig. 5b) supporting a link between the oxic water CH₄ production and algal-derived organic matter substrate availability⁹."*

L328-333 This sentence is pretty dense. I'd expand this part to detail your calculations better, and walk the readers through this. Please see next comment

L328-333 Also worth reminding readers that your calculated oxic production does agree well with previous experimental estimates of production, which further supports your conclusions.

This part is now slightly different (L369-374):

"In Lake Hallwil's metalimnion, we assess that lateral transport accounts for $10 \pm 1 \text{ nmol CH}_4 \text{ L}^{-1} \text{ d}^{-1}$ (Table 1) leading to an accumulation of $\sim 5.0 \pm 5 \text{ nmol L}^{-1} \text{ d}^{-1}$ (Fig. 6) of which an average of $0.3 \pm 3 \text{ nmol L}^{-1} \text{ d}^{-1}$ is locally produced/consumed. Contrarily, in the surface mixed layer we estimate a significant and

unaccounted for internal source of CH₄ (P_{net,sr}, Fig. 1) of 110 ± 60 L⁻¹ d⁻¹ that is in the same range of what was estimated by laboratory and mesocosm-based studies^{9,13}.“

L348 Worth comparing the 120 mol d⁻¹ to values in Table 2 – this number is very small compared to surface oxic production rates.

We agree and now expanded this part (lines 382-390) to:

“The maximum input was estimated as 120 mol d⁻¹, which represent only 3 to 6 % of the estimated P_{net,s}. However, even such contribution to the mass balance is very conservative. In fact, if the plume was carrying CH₄ upward, then we would see elevated concentrations below the thermocline in the area of detrainment. The main sampling station (A on Supplementary Fig. 2) is only about 250 meters south of the diffuser ring. Here the CH₄ profiles show that water at 40 m depth was near the detection level of the method in all cases (Fig. 4), indicating the concentrations within the plume itself are likely near this background concentration.”

L364 Could cite Winder & Schindler 2004 Glob Chg Biol as evidence supporting this lengthened stability period.

We added the citation as reference , L392 (n.63).

L368 ‘System analysis’ could be changed to ‘detailed whole lake mass balancing combined with incubation and isotopic approaches to show...

We replaced “system analysis” with “detailed whole lake mass balancing combined with incubation and isotopic approaches to show...” Now line 398.

L386 change ‘shaded’ to ‘sheltered’. We changed accordingly (L420).

L415 What do you mean by ‘longitudinal transect’ .

We added “(center-south)” to clarify this (L451).

L463 Was -36 per mil at one depth in particular or across the dataset?

Here, to be more precise, we add that the isotope values refer to incubations carried out at 8 m depth. (L509).

L464 Measured values in the lake?

We add to line 509 “measured for the incubations at 8 m depth.”

L506 and L 532 I’d refer to figure 1 throughout these parts to help readers follow

We now added “Figure 1” to each of the mass balance components (L583-585).

L557-569 unclear in this section what method was ultimately used in the mass balance. It is worth stating explicitly how this term was estimated.

L618-621 read now as: "Average flux (April – August 2016) is equal to $0.8 \pm 0.2 \text{ mmol m}^{-2} \text{ d}^{-1}$ from McIntyre relationship for positive buoyancy and to $0.6 \pm 0.3 \text{ mmol m}^{-2} \text{ d}^{-1}$ from chamber measurements. The latter, not significantly different from the wind based relationship, was used for the mass balance (Table 2)."

L572 What is a localized durable hot-spot? How was it determined?

According to previous study from Flury et al. (2010) the most (methane) productive areas in Lake Hallwil are those next to the reeds. We determined diffusive fluxes for these same highly productive areas defined as durable hot-spots. For clarity we replace this definition with "*Diffusion from the sediment to the water column was estimated at shallow sites characterized by reed vegetation*" L624-625.

L579 why is this conservative?

As to L 626-631: "The contribution to dissolved methane by ebullition was estimated from ebullition rates determined at the same sites by Flury et al.(2010) assuming bubbles entirely composed by methane and that all of the bubble methane dissolve into the water ($1.2 \pm 0.8 \text{ mmol m}^{-2} \text{ d}^{-1}$). For mass balance purposes, the total littoral CH_4 flux to the water column was conservatively assumed to be emitted from the whole lake sediments surface between 0 and 10 m depth."

This is a conservative estimate because the highly productive areas are assumed to cover entirely lake sediments between 0-10 m depth, while it was estimated by Flury et al., that these areas correspond to less than 2% of the lake planar area.

L613 List symbol for this depth dependent production. This is now correct.

Display Items:

Fig. 1 – Define F_r from the river in legend. F_r is defined in the figure caption.

Fig 4 – legend: no CH_4 oxidation, or just a little based on result in line 117. Why are surface MOx rates not in the bar plots?

We added Mox rates to plots in Figure 4, specifying where isotopes did not show any change.

Fig 6 – I'd remove CH_4 concentration profiles, they clutter the graph and take away from the main result.

We agree and now removed the CH_4 profiles from Figure 6.

Why is the figure not extended to the surface layer? Can you define P and R more clearly in the legend?

As explained in lines 176-182 Fick's 2nd law (Eq. 2) is not applicable to the surface mixed layer where the water column is not stably stratified. As explained in line 635 we resolved vertical diffusivities (K_z) with the heat budget method for the stratified water below 6 m (Powell and Jassby 1974). The diffusivity estimate is based on the assumption that at depth z , the vertical turbulent transport of heat is equal to the rate of change of heat content below that depth. Consequently, this method is not applicable above 6 m (SML), where surface heat gain/losses (solar, cooling, wind, etc.) generate convection and therefore cannot be used to infer turbulent diffusion.

As to Reviewers comment on P and R, following the suggestion in the general comments, we replaced these terms with P_{net} and P_{gross} (as well in the Figure 1 legend).

Fig S2 – the range is unclear in legend, why are the profiles not for a specific date?

The method involves finding the increase in temperature over a given time, and thus is usually calculated on ~monthly data, or periods where there are obvious changes in warming rates (as we have done).

Fig S5 – Units on axes? The figure was modified accordingly. Please note that this is now Supplementary Fig. 6.

Table S1 – do you mean R, not R^2 ?

The reported values are R^2 .

Reviewer #2 (Remarks to the Author):

The methane paradox is the notion that methane is produced in oxic surface waters by de-methylating processes when seemingly it should be a strictly anaerobic process. I don't doubt that aerobic methane production occurs. But could it be 80% of the methane produced at this study's lake? Despite an admirable argument built by these authors, I remain unconvinced. First, I think that the authors' raise a high bar for themselves. In line 32 they say that the 'methane paradox is an increasingly controversial topic.' Then on line 65, they state that they have "come to the unprecedented conclusion that most of the CH₄ production takes place in the" fully aerobic "surface mixed layer." This sets the stage for a totally convincing documentation of the process, which while the authors do a fine job of inference, they fail to convince.

Thank you for the comments. Of course, we did search for additional sources to explain the > 80%, particularly methane ebullition. In fact, we also approached our mass balance in a very conservative manner, using high end values and estimates for the various inputs. We have added some additional text which we are optimistic that the reviewer will find convincing (L80- 83; 221-223; 267-270; 375-390).

We are confident that ebullition and the bubble plume diffuser play negligible roles in supplying methane to the surface water, as we will explain below. In fact, methane ebullition never developed in the hypolimnion of Lake Hallwil and hypolimnetic methane concentrations remain close to detection levels. These arguments will be detailed in our responses below to the specific concerns of the review. Please note that after uncertainties assessment (Table 1 and 2, L656-664 and Supplementary Table 2) the overall contribution of surface CH₄ production to the evasion to atmosphere is estimated >80%.

1. In their conceptual model of the lakes methane “system” the author set the stage by portraying the metalimnion as a methane maximum overlain by a methane poor epilimnion. Most of the methane production is thought to occur in this surface layer, while the metalimnion is a place where methane kind of gets stuck. High concentrations are not indicative of high rates of production. With this I agree.

Thank you for the comment.

2. Evidence that bulk of the methane is produced in the surface mixed layer, the epilimnion, is inferred by mass balance, coupled with direct measurements of methane oxidation. The relatively high rates of surface emission cannot be supported by vertical flux across the metalimnion or from the littoral zone so therefore they must be produced in the oxic surface waters (Table 2).

This is correct.

3. While there is certainly a host of detailed interesting measurements in the paper, a couple of things trouble me, mostly focusing on bubble transport and the entrainment of water by this process. In figure 1, the bubbles in the center of the lake are transported from the “diffuser.” One is extremely puzzled by this term and doesn’t know what to do with it, until perusing the supplemental material hours later, one discovers that the lake is sparged from below. We are assured, however that the sparging is only to prevent anoxia in the deep lake waters, and that it doesn’t upset the stratification of the lake itself.

We agree with this, and now introduce the diffuser term much sooner (lines 78-83), and reshape Figure 1 to better represent the term F_D (i.e. potential diffuser-driven transport from the hypolimnion). The bubbles rising from the bottom in the previous drawing meant to represent the diffuser, but we realize it may mislead the reader and be interpreted as a methane ebullition from deep sediments, which, as we describe below, does not exist in Lake Hallwil. We hope figure in this form will avoid misunderstanding. The diffuser system and resulting plume was shown previously that it does not impact the stratification (see McGinnis et al. Referenced in the following paragraph). The aeration system operation has since been drastically reduced, and now runs 12 hours a day only with air at 100 Nm³/h.

Figure 1: Conceptual schematic of the CH₄ budget in mesotrophic Lake Hallwil. CH₄ mass balance components: evasion to the atmosphere (F_s), interior turbulent diffusion (F_z), transport from the diffuser (F_D), lateral transport (F_L) and river input rate ($F_R = Q_R \times C_R$). The case study (Lake Hallwil, Switzerland) was divided into zone 1 (metalimnion) and zone 2 (surface mixed layer). The mass balance reveals that average dissolved CH₄ concentration in the summer shows an CH₄ metalimnetic maximum concentration (zone 1), however with low production rates ($P_{net,m}$). The highest CH₄ production rates ($P_{net,s}$) are actually at the surface (zone 2). The turbulent exchange at the lake surface act to mitigate the zone 2 CH₄ concentrations by enhancing outgassing, while the metalimnion gas exchanges are driven by turbulent diffusion.

I was troubled by this sparging for two reasons. If the deep lake has so much oxygen demand that it requires sparging to not go anoxic, wouldn't one expect greater rates of methane production in its sediments,

Methane production indeed occurs in the sediment (as evident in Fig. 5a), but not at rates high enough to support ebullition. We have several strong arguments to support this conclusion:

1. Retrieved sediment cores below the thermocline show no signs of ebullition or gas voids (figure below). The core images from June 2015 (own data, now added to Supplementary Information) and November 2011 (courtesy of Dr. Beat Müller) show no signs of active ebullition.

2. Porewater methane concentrations remain below saturation values. For example, at 15 m and 10°C, methane porewater concentrations would have to exceed $\sim 4.5 \text{ mmol L}^{-1}$ before they would become oversaturated (including dissolved N_2) and bubbles would begin to form. This value increases to $\sim 8.5 \text{ mmol L}^{-1}$ at 30 m depth and 5°C, and 12 mmol L^{-1} at 45 meters depth. The maximum porewater concentrations measured were 1.5 mmol L^{-1} at 7.7 m, 3.7 mmol L^{-1} at 23 m, 5.9 mmol L^{-1} at 29.3 m and 2.0 mmol L^{-1} at 45 m depths. These are all below the saturation concentration and thus bubbles will not be formed.

B [Redacted].

Figure 1 A) Picture of top 20 cm of a core section taken at 46 m depth in Lake Hallwil (St. A) on 12 June 2015 – now added as Supplementary Figure 1. B) [Redacted].

3. We further argue the following: a) the thermocline mass balance is very well-constrained. If bubbles were released below that would contribute to the surface methane, then these would add proportionately more methane to the thermocline than to the surface as the dissolution rate would be higher due to a higher concentration driving force (McGinnis et al. 2006). Tang et al. (2014) performed a similar analysis with bubble modeling and showed that the bubbles released below the thermocline would mostly be dissolved before reaching the thermocline and surface water. Methane bubbles in natural waters have ranges generally between 4 – 6 mm in diameter (McGinnis et al. 2006, DelSontro et al. 2015).

However, Lake Hallwil has no free gas in the sediment below the thermocline as shown by our numerous sediment coring and very low bottom concentrations.

The same is true with the diffuser. If it was transporting significant methane, this would be mostly apparent in the bottom water where the plume detains. As our main sampling station was about 250 meters away from the diffuser system, we would have seen a signal in the methane concentrations.

5. As a sensitivity analysis, we can evaluate how much bubbling in the shallow regions would be necessary if it would contribute to entire 80% of the missing methane to the surface water (0 -5 m).

The surface volume is 0.04 km^3 and the sediment area is 0.1 km^2 covering 0 – 5 m. The production is $\sim 70 \text{ kg/d}$. This would imply we would need a contribution of $50 \text{ mmol m}^{-2} \text{ d}^{-1}$ to the surface mixed layer by dissolving bubbles. We model that an average 4 - 5 mm bubble (see McGinnis et al. 2006) released from 5 m depth would dissolve about 20% in the surface water before reaching the atmosphere. To supply the $50 \text{ mmol m}^{-2} \text{ d}^{-1}$ would imply a methane bubble flux of $250 \text{ mmol m}^{-2} \text{ d}^{-1}$ from these regions, with about $200 \text{ mmol m}^{-2} \text{ d}^{-1}$ then reaching the atmosphere. This would be beyond the very upper range of very high-ebullition rates recorded in hydropower reservoirs ($<100 \text{ mmol m}^{-2} \text{ d}^{-1}$, Deemer et al. 2016), and imply a lake emission of 400 kg d^{-1} , 5 times higher than we report.

Regarding the diffuser: The aeration system was put into place in 1986 to combat anoxia and support the local fishery. This early intervention, combined with aggressive nutrient management, seemed to stop the eutrophication before ebullition could develop. At present, the hypolimnion remains oxic through the entire stratified season, except that the bottom 5 meters became anoxic in October 2015, and Sept 2016.

We now added to the text to better explain the negligible role of ebullition from deep sediments and the presence of the aeration system:

L75-83: "After rigorous restoration measures for the past 30 years, the lake reached a mesotrophic state in 2008 (see Supplementary Note). The re-oligotrophication process was supported since 1986 by the installation of a hypolimnetic aeration system, placed on the lake bed at ~46 m depth (see Supplementary Note). The air/oxygen flow rate of the system is regulated such that, while preventing anoxic conditions in the deep water, the rising bubble plume does not affect the stratification of the water column in summer²². The aggressive restoration measures were extremely effective, resulting in a re-oligotrophication of the lake where bottom waters remain near completely oxic and preventing methane ebullition from developing in the hypolimnetic sediment (Supplementary Fig. 1)."

L375: "Low CH₄ concentrations (< 0.05 μmol L⁻¹) between 25 and 45 m depth at the lake center led to the conclusion that any CH₄ diffusing from deep sediments (5 and 6 mmol m⁻² d⁻¹ at 23 and 45 m, respectively) does not reach the metalimnion or the SML. The highest oxidation rates are likely taking place within the surface sediments as reported for other studies in mesotrophic lakes^{5,62}. However, the presence of the aeration system may favor the transport of bottom water CH₄ within rising bubbles. This contribution to the bulk CH₄ content at 5 – 10 m depth was quantified using modeling of air bubbles at equilibrium with the highest measured bottom water (46.5 m) CH₄ concentration (7 μmol L⁻¹, see Methods). The maximum input to the thermocline (zone 1) was estimated as 120 mol d⁻¹, which represent only 3 to 6 % of the estimated P_{net,s}."

and 2) is it not possible that this sparging would set up localized plumes of water advecting upwards carrying methane into the mixed surface layer without upsetting the overall lakes stratification?

In short, no this would not be possible without disrupting the thermocline. If this would occur, then it would imply a very high production in the thermocline similar to what we reported for the surface layer (as the transported methane from the thermocline would have to be replaced). The bottom water simply does not have much methane to transport.

As we mentioned, the balance we did on the diffuser system was very conservative. In fact, if the plume was carrying methane upward, then we would see elevated concentrations below the thermocline in the area of detrainment. The bottom water was near the detection level of the method in all cases indicating the plume itself is likely near the background concentrations levels of the bottom water (Fig. 4). The main sampling station, A on Supplementary Fig. 2, is only about 250 meters south of the diffuser ring, so this would have been evident in our methane profiles. If the plume was transporting methane-enriched water somehow through the thermocline, then this would also be apparent in the thermocline methane – though we did not see any lateral signal that would suggest this was so (Fig. 3).

To clarify this issue, we the following text:

L384-390 " ... even such contribution to the mass balance is very conservative. In fact, if the plume was transporting CH_4 from the benthic boundary layer upward, then we would see elevated concentrations below the thermocline in the area of plume detrainment (the main sampling station A, on Supplementary Fig. 2, is only ~250 meters south of the aeration system diffuser ring). Here the profiles show that dissolved CH_4 below the thermocline to ~40 m depth was near the detection level of the method in all cases (Fig. 4), indicating the concentrations within the plume itself are likely near this background concentration."

We made a simple calculation to investigate the "mini-plume" theory and assumed that the plume was enhancing the local diffusivity between Zone A and Zone B. To summarize, K_z would have to be $1 \text{ m}^2/\text{s}$ at the plume area to support the transport of methane to the surface layer, 1,000,000 times higher than we measured. From an energetic perspective, the dissipation of turbulent kinetic energy (ϵ) would then have to be $2.7\text{E-}2 \text{ W/kg}$. Maximum ϵ in lakes is typically $1\text{E-}7 \text{ W/kg}$ in the surface layer and bottom boundary layers (Wüest et al. 2000). Spread over the area of the diffuser, if this was occurring, severe mixing would be seen in the temperature structure, as well as from our turbulence measurements using the heat budget method (minimum measured diffusivity $K_z \sim 1\text{E-}6 \text{ m}^2/\text{s}$). Weighted averaging over the entire basin would increase the basin-scale diffusivity from $1\text{E-}6$ to $7\text{E-}3 \text{ m}^2/\text{s}$, a 7000 times increase that would certainly disrupt the thermocline, and be evident in our measurements.

4. Figure 4, continuing the thought above, sets up the idea in the two upper panels that the methane maximum is confined to the metalimnion, centered around 5-7 meters depth, but figure 4 shows significant concentration all the way to the surface, nearly 80% in June and 55% in August of the

concentrations of methane at the surface relative to the metalimnion. Could there not be a mechanism for bypassing the stratified layer bringing methane into the surface mixed layer and supporting the high flux from the surface?

We searched for and subsequently ruled out all possibilities we could consider. Based on the above calculations, such a bypass would be enormous. If they existed, they would be evident and alter the thermal structure. We measured all inflows, including the sewage treatment plant (it has no methane and a deep discharge).

The important thing to note is that we propose that the surface concentrations are almost entirely decoupled from the thermocline. The source appears relatively constant in that the values did not change much in the SML despite the relatively high flux to the atmosphere.

5. Overall, I think ebullition from the sediments is under represented in this model. While great detail is given on limnology measurements (391-410, water column measurements (412-441), surface methane flux 541-570) the contribution of ebullition is dismissed in three lines 575-577 and referenced away. I don't see that ebullition from deeper zones is considered, despite the high oxygen demand in the deeper part of the lake. Ebullition has found to be important in many lakes even those not eutrophic or thermakarst. The bubbling rates used by the authors, Flury et al (in ref list) were made on the shores of this lake in a marsh dominated by Phragmites. Emergent aquatic macrophytes are known to suppress ebullition. In the unvegetated portions of the lake shore, ebullition should be higher.

Wik, M., Crill, P. M., Varner, R. K., & Bastviken, D. (2013). Multiyear measurements of ebullitive methane flux from three subarctic lakes. *Journal of Geophysical Research: Biogeosciences*, 118(3), 1307-1321.

Wik, M., Thornton, B. F., Bastviken, D., MacIntyre, S., Varner, R. K., & Crill, P. M. (2014). Energy input is primary controller of methane bubbling in subarctic lakes. *Geophysical Research Letters*, 41(2), 555-560.

I don't suggest that ebullition is supporting surface fluxes, but ebullition, higher lateral methane transport, and sparging of deeper waters may be bringing methane to the surface mixed layer.

See our above comments. We actually feel that the Flury et. al (2010) rates applied to the whole lake is very conservative, in fact, we end up applying the rates reported by Flury et al. 2010 as if the CH₄ (assuming 100% bubble content) contained in bubbles is all dissolved into the water column (and homogeneously along the lake surface). As we show in the above estimates, the bubble rates would have

to approach the worse bubbling hydropower to account for our fluxes. We think that we convincingly argued that there are negligible deep water bubbling contributions.

To clarify this, we state: L221-223: *“This is a conservative estimate, as it was assumed that the whole sediment surface of the lake from 0-5 meters is subject to highest-rates of ebullition ($1.2 \pm 0.8 \text{ mmol CH}_4 \text{ m}^{-2} \text{ d}^{-1}$ dissolved in water, representing 1.2 % of the lake area according to Flury et al. 2010).”*

6. Incubations. The authors conducted incubations to constrain methane oxidation rates. The results are very interesting showing higher incubation rates in the deep waters where ^{13}C is enriched. I wondered though, the incubations of surface water showed very little change in methane concentration, and these were interpreted that oxidation rates in the surface mixed layer are nil. However, if methane production is so important in this surface mixed layer, why didn't there incubations show increases in methane concentration over time? Maybe the mechanism doesn't work in a bottle? In that case, if the authors want to support their “unprecedented conclusion” about such a controversial topic, in my opinion a direct measurement of methane production would convince me.

We agree with Reviewer 2 that the absence of CH_4 production rates in our *in situ* incubation is puzzling. After ~1 month (e.g. July-August) we observed a decrease in CH_4 in 120 mL serum bottles at 2, 8, and 13 m. However, only the measurement at 13 m depth is supported by the concomitant $\text{d}^{13}\text{C-CH}_4$ enrichment (=20 ‰), confirming that the CH_4 decrease is ascribable to methanotrophes. On the contrary, the hypothetical oxidation rate of $\sim 3 \text{ nmol L}^{-1} \text{ d}^{-1}$ measured at 2 and 8 m depth is associated to a constant isotope value for methane. This may indicate a production that adds very low isotope values to the medium.

We agree with the Reviewer that a production of $100 \text{ nmol L}^{-1} \text{ d}^{-1}$ should have been more than evident during these incubations, even with oxidation occurring. Indeed, the reason why we did not observe production rates in the bottles remains unanswered and, as the Reviewer suggests the bottle may exclude some important mechanism. This is the subject of future studies. However, the mass balance remains convincing.

On this point, we would like to point out how there are a number highly qualified research groups that are working on the complex pathways that may lead to “oxic methane production”. This is however a complex issue, since different aquatic systems may experience a variable contribution of several “oxic methane production” mechanism (perhaps related to algal metabolism, methylphosphonates of terrestrial origin, dissolved organic matter, zooplankton interaction, UV interaction, etc). Even in the seminal study by Grossart et al. (2011), the surface water incubations showed no methane increase.

Furthermore, for them methane production in dark incubation was significantly higher than production during light incubation, while we hypothesize a link with light penetration. That is to say, there are multiple questions remained to be answered. As such, **the main message of our work to the scientific community is that methane production in oxic lake water at the (mostly reported) metalimnetic CH₄ concentration maxima represents 1% of the rates occurring at the surface. Thus efforts on further experiments should take this into account.** We are confident that this will promote advances on the definition of pathways behind the methane paradox.

A compelling study on the topic of methane oxic production and MPn by Wang et al. (accepted by *Environ. Microbiol.* 1–41, 2017) showed that their sampling efforts strictly addressed the metalimnetic maximum CH₄ concentration (nothing reported above 10 m depth). Although some evidence was produced, we believe their work represents a missed opportunity at also considering the production in the surface layer.

In conclusion we are thankful for the comment as we implemented the results and discussion with some of these aspects that were previously not included, and that we believe has improved the manuscript.

We now add MOx rates and isotope data to Figure 4 and the following text:

L265-273: *“In the SML of Lake Hallwil the situation is vastly different. We show that during the stratified season, the most significant production rate ($P_{net,s} = 110 \pm 60 \text{ nmol L}^{-1} \text{ d}^{-1}$, Fig. 1) is mostly expressed in these upper 5 m and not in the metalimnion. Bottle incubations in the SML show either negligible CH₄ oxidation ($0.3 \text{ nmol L}^{-1} \text{ d}^{-1}$, June-July 2016) or higher oxidation ($3.6 \pm 0.2 \text{ nmol L}^{-1} \text{ d}^{-1}$, July-August 2016) with no change in isotope values (2‰ after 1-month incubation). **This may indicate oxidation is compensated by a CH₄ production mechanism.** The magnitude of the surface CH₄ production is however masked by the relatively rapid water-air exchange. As a result of the CH₄ loss to the atmosphere, the observed CH₄ concentrations remain lower and fairly consistent in the surface layer versus the metalimnion.”*

L307-311: *“With a $\delta^{13}\text{C}_{\text{CH}_4}$ equal to -62‰ at Time 1, as from in situ measurements, the isotope value associated to the measured consumption rates of $3 \text{ nmol L}^{-1} \text{ d}^{-1}$ should have been in the order of -41‰ instead of the final measured -64‰ (assuming MOx with fractionation factor of 20‰). **This might suggest a local “compensation” with low isotope values.**”*

L341-345: *“Only recently, methyl-phosphonate (MPn) biodegradation has been indicated as a possible source of methane in oceans, a theory which was confirmed to apply to mesotrophic lakes by a recent work on Lake Yellowstone. However, **in the mentioned study, and contrary to our findings, laboratory efforts focused on samples taken at the metalimnetic methane peak that occurs during stratification.**”*

7. Isotopes. The lighter isotopes in the surface and enriched ones in the deeper lake waters are consistent with the oxidation pattern measured by the authors. However the lighter surface mixed layer $\delta^{13}\text{C}$ is also consistent with transport from sediments via ebullition. Production of methane from methylated substrates often leads to enriched methane as the substrates are limiting and used to completion. Eg. Kelley, C. A., Poole, J. A., Tazaz, A. M., Chanton, J. P., & Bebout, B. M. (2012). Substrate limitation for methanogenesis in hypersaline environments. *Astrobiology*, 12(2), 89-97.

We appreciate Reviewer 2 addition to our literature review. We actually observe that, if organic matter was the CH_4 precursor in the SML, its depletion would have led to a CH_4 with higher isotope values (or enriched) as compared to methanogenesis in porewater. However, as we say in L325, we do not know the isotope value of possible precursors as methylated compounds - which depends strongly on formation pathways. Finally, we cannot exclude any isotope value until we have more insights on the process(es) involved.

We clarify this in the following text:

L245-250: "Sediment CH_4 production of Lake Hallwil exhibits an α_{app} of 1.056 – 1.060, which is characteristic for environments dominated by acetate-dependent methanogenesis. Estimates for the water column SML methane production show a smaller fractionation factor ($\alpha_{\text{app}} = 1.045$). Consequently, in Lake Hallwil we observed a different isotopic fractionation between the CH_4 produced in sediments and in the SML, however both characteristic for acetoclastic methanogenesis."

L322-327. "Yet the carbon isotope composition of CH_4 can be influenced by the type of acetate precursor, the production mechanism(s) and pathways, and relatively little is known about methanogenesis of oligotrophic lake surface sediments. Lake Hallwil apparent fractionation (1.056 – 1.060) of sediment CH_4 production indicates an acetate-dependent methanogenesis, which is in good agreement with temperate, oligotrophic lake sediments (e.g. 1.065, Lake Stechlin)."

L337-340: "However, the estimates of fractionation factors performed here, as in the majority of methanogenesis studies, are based on the assumption that there is no major methanogenic precursor other than acetate or CO_2 . Arguably, additional and perhaps novel pathways should be evaluated."

References not present in the manuscript

DelSontro, T., McGinnis, D.F., Wehrli, B. and I. Ostrovsky. Size Does Matter: Importance of Large Bubbles and Small-Scale Hot Spots for Methane Transport *Environmental Science & Technology* 2015 49 (3), 1268-1276 DOI: 10.1021/es5054286

Deemer B.R. et al., Greenhouse Gas Emissions from Reservoir Water Surfaces: A New Global Synthesis
BioScience 66(11). 2016. DOI: 10.1093/biosci/biw117

Wüest, A., Piepke, G., Van Senden, D. C., Turbulent kinetic energy balance as a tool for estimating vertical diffusivity in wind-forced stratified waters. *Limnol. Oceanogr.* 2000, 45, (6), 1388-1400.

Reviewer #3 (Remarks to the Author):

Several recent studies provided strong evidence of methane production in oxygenated freshwaters challenging the long-standing paradigm that microbial methane production occurs only under anoxic conditions and forces us to rethink the ecology and environmental dynamics of this powerful greenhouse gas. Conventional explanations for this paradox include input from nearby anoxic sediments and shorelines and production within microanoxic zones in deeper layers of lakes (metalimnion). However, the study by Donis and co-authors provide unambiguous evidence for methane production within the surface mixed layer (epilimnion) under high oxygen concentrations. Thus methane production in the upper oxic water layers places the methane source closer to the air water-interface, where convective mixing and microbubble detrainment might lead to a methane efflux higher than that previously assumed.

Therefore the reported results of this study are novel, highly interesting and important as they further improve our understanding of aerobic methane formation in surface waters of lakes. I strongly favor the publication of this work in Nature Communications. However, to improve the work I have minor comments listed below which the authors should pay attention to.

Abstract, line 19-20 "...we reveal that 80 % of CH₄ emissions to the atmosphere (>80 kg d⁻¹) are due to CH₄ produced within the lake surface mixed layer...".

This sentence implies that methane emissions are occurring frequently at similar rates throughout the whole year. I presume that the calculated daily average rate of CH₄ emissions and the total amount (~20 tons yr⁻¹) is only valid for the season from May until October. Please revise sentence in the abstract and clarify this issue throughout other parts of the manuscript (results and discussion).

We agree with the Reviewer and changed the following parts accordingly.

L 18-21 "Our mass balance on a temperate, mesotrophic lake revealed that ~90% of local CH₄ emissions to the atmosphere are due to CH₄ produced within the oxic surface mixed layer (SML) during the stratified period, while the often-observed CH₄ maximum at the thermocline represents only a physically-driven accumulation"

L147-150: "Sources and sinks **during the lake stratified period** (May – October) of 2016 were determined dividing the surface layer in two key zones as shown in Fig. 1."

L218: "...revealing a source of CH₄ ($P_{net,s}$ in eq. 3) of $110 \pm 60 \text{ nmol L}^{-1} \text{ d}^{-1}$ ($73 \pm 40 \text{ kg of CH}_4 \text{ per day during stratified periods}$)."

L265: "In the SML of Lake Hallwil the situation is vastly different. We show that **during the stratified season**, the most significant production rate ($P_{net,s} = 110 \pm 60 \text{ nmol L}^{-1} \text{ d}^{-1}$, Fig.1) is mostly expressed..."

L398-400: "In the present study we used detailed whole lake mass balancing combined with incubation and isotopic approaches to show that in Lake Hallwil, **during the stratified period**, up to 90% of the emissions ($25 \pm 18 \text{ tons yr}^{-1}$) result from local methane production"

Abstract, line 22 "Fractionation between the CH₄ precursors and CH₄ in the surface mixed layer is distinct from sedimentary methane production (30 ‰ vs. 40 ‰)."

Sentence is unclear (isotope terminology!), better write "Stable carbon isotope values indicate that CH₄ in the surface mixed layer is distinct from sedimentary CH₄ production".

We rephrased the sentence as suggested (L23-24): "Stable carbon isotope values indicate that CH₄ in the SML is distinct from sedimentary CH₄ production, suggesting distinct pathways and precursors".

Page 2, line 47. "... (2) algal metabolites..."

This part needs slight modification as the study by Lenhart et al. has shown that the marine algae *Emiliana huxleyi* is able to produce CH₄ *per se* but also from the thio-methyl group of added methionine. In this context I would like to inform the authors about a just published paper that describes a chemical mechanism for iron-oxo-catalyzed methane formation (Comba, P., Benzing, K., Martin, B., Pokrandt, B. and Keppler, F. ()), Nonheme iron-oxo-catalyzed methane formation from methyl thioethers: Scope, mechanism and relevance for natural systems. Chem. Eur. J.. Accepted Author Manuscript. doi:10.1002/chem.201701986).

We thank the reviewer for the clarification. The sentence reads now as (L44-46): "Other postulations for the presence of CH₄ in oxygenated waters include: (1) anoxic micro-niches^{12,13} (2) algal metabolites, with methionine as a possible precursor¹⁴, and (3) CH₄ as a by-product of methylphosphonate decomposition¹⁵"

Page 5, Figure 2, unit given for chl *a* in legend and figure is not consistent (μmol L⁻¹ or μg L⁻¹ ?)

Thank you for the remark, we have added the correct units to the legend.

Page 15, Table 3. Replace "Total depletion" by "isotope difference between $\delta^{13}\text{C}_{\text{CH}_4}$ and $\delta^{13}\text{C}_{\text{DOC}} / \delta^{13}\text{C}_{\text{TOC}}$ ". Values should be negative. Also rewrite sentence in the manuscript at line 282-284.

We replace total depletion in the table with " $\delta^{13}\text{C}_{\text{DOC,TOC}} - \delta^{13}\text{C}_{\text{CH}_4}$ "

And rephrased as suggested (now L233-235): "*We infer a smaller difference between the isotope values of carbon source and CH_4 produced in the oxic water column (-32 – -29 ‰) as compared to the sediment methanogenesis (-44 – -41 ‰, Table 3).*"

Statistics: For almost all Figures, Tables and calculations in the text regarding methane fluxes (e.g. daily and total flux) and mass balances from different compartments no statistical values were provided (e.g. error bars, SDs, uncertainty range, number of replicates, etc.). Furthermore, no information about the analytical uncertainties for the measurement systems described in the method section was provided (e.g. stable isotope measurements). This needs to be substantially improved in the revised manuscript.

We agree with the reviewer, and have now greatly strengthened this aspect of the manuscript.

We have assessed the uncertainties in the mass balance(s) that now appear in Table 1 and 2.

To assess overall uncertainties in the mass balance(s), we ran Monte Carlo simulations (999 iterations), in which each component in the calculation is randomly picked from a normal distribution around its mean value ($F_s, F_L, F_z, P_{\text{gross,m}}$), or from a range going from 0 to a maximum estimate (F_R, F_D). – Now added to the Methods, L656.

The following Table was added to the supplemental material (Supplementary Table 2). The table summarizes mass balance(s) components mean and standard deviations (s.d.) as well as the number of samples and the time (time range) when measurements were performed.

Supplementary Table 2

Mass balance component	Mean/range	s.d., N, unit	Measurement
F_s (Evasion from surface)	0.6	0.3 (28) $\text{mmol m}^{-2} \text{d}^{-1}$	April-Aug 2016
F_L (sed) (Diffusion from littoral sediments)	1.75	0.2 (2) $\text{mmol m}^{-2} \text{d}^{-1}$	Sept 2016
F_L (eb) (Dissolution from littoral ebullition)	1.2	0.8 (8) $\text{mmol m}^{-2} \text{d}^{-1}$	Flury et al. 2010
F_z (Diffusion from metalimnion peak)	0.03	0.01 (3) $\text{mmol m}^{-2} \text{d}^{-1}$	June-August 2016
F_D (surface CH_4 contributions from the aeration system)	0-0.014	$\text{mmol m}^{-2} \text{d}^{-1}$	
F_R (Input from rivers)	0-0.005	$\text{nmol L}^{-1} \text{d}^{-1}$	
$P_{\text{gross-m}}$ (Production from Eq. 2)	0.01	0.001 $\text{nmol L}^{-1} \text{d}^{-1}$	

By this range of uncertainties, we determined a plausible $P_{\text{net},s}$ deviation from the mean ($110 \pm 60 \text{ nmol L}^{-1} \text{ d}^{-1}$), that is largely ascribable to the air-water flux s.d.

Note:

- F_D (surface CH_4 contributions from the aeration system): the upper end of the range describes gas transfer across the surface of an individual rising bubble with the flow rate of the aeration system ($180 \text{ Nm}^3 \text{ h}^{-1}$) assuming a bubble diameter of 4 mm and a bottom CH_4 concentration of $7 \mu\text{mol L}^{-1}$ (maximum measured at 46.5 m).

- F_R (input from rivers): the upper end of the range corresponds to input of CH_4 from rivers, estimated by the product of the flow rate ($Q_R = 2.5 \text{ m}^3 \text{ s}^{-1}$ for rivers and $1 \text{ m}^3 \text{ s}^{-1}$ as conservative average for periodical surface runoff) and the maximum CH_4 measured in front of the Aabach river mouth ($1 \mu\text{mol L}^{-1}$), corrected for background surface lake CH_4 of $0.3 \mu\text{mol L}^{-1}$.

- $P_{\text{gross},m}$ (gross methane production obtained in the metalimnion between 5-10 meters): standard deviation was set as 10 %. i.e. the coefficient of variation of CH_4 concentration profile measurements. To this point we add to the Methods the following:

L479-484 “Each sample procedure from the Niskin bottle to the gas bag takes ~10 minutes. To prioritize a higher vertical resolution, given the time consuming procedure (a longer time would increase the uncertainty linked to the natural variability of the water column structure) we performed replicates only once and applied the determined coefficient of variation (averaging 10.0 % for CH_4 and 10.6 % for CO_2 over a set of 8 replicates) to the data set as a \pm range of uncertainty of the measurement.”

As to the analytical uncertainties for the measurement systems, we now add to the Methods section the following information:

L471-476:” Cavity Ring-Down Spectrometer analyzer (Picarro G2201-i, Santa Clara, CA, USA) for immediate reading of concentrations in the gas phase (ppm) and stable isotope ratio ($\delta^{13}\text{C}$ in ‰ vs VPDB standard). Instrument specific precision at ambient concentrations (1- σ of 5 min average) is < 0.16 ‰ for $\delta^{13}\text{C}$ in CO_2 , < 1.15 ‰ for $\delta^{13}\text{C}$ in CH_4 , for [$^{12}\text{CO}_2$] is 200 ppb + 0.05 % of reading and for [$^{12}\text{CH}_4$] is 5 ppb + 0.05 % of reading.”

L543 “The precision of the $\delta^{13}\text{C}_{\text{DOC}}$ measurements by EA/IRMS was better than 0.1 ‰”

L550 “The precision of the $\delta^{13}\text{C}_{\text{TOC}}$ measurements by EA/IRMS was better than 0.05 ‰. “

L589-591:” portable GHG analyzer (UGGA; Los Gatos Research, Inc.) Instrument specific precision at ambient concentrations (1- σ of 100 s average) for [$^{12}\text{CO}_2$] is 40 and for [$^{12}\text{CH}_4$] is 0.25 ppb. “

L639-643:” The loggers (RBR TR1060, Ottawa, Canada) at 5, 9 and 11.5 meters measured temperature every 5 seconds with a 0.1 second response time and 5×10^{-5} °C resolution. The remaining loggers (Vemco Minilog-II-T loggers, Canada) were recording every 1 minute, with a resolution of 1×10^{-2} °C and response time of < 5 minutes.”

Reviewers' Comments:

Reviewer #1:

Remarks to the Author:

The authors have put in a great deal of effort to address reviewer concerns, and I think the manuscript is in excellent shape. I only have a few minor comments:

L64. The sentence is a bit awkward, I would rephrase to something like "... is in direct contact with the atmosphere, implying that lake surface waters may be an important, but overlooked site of CH₄ production".

L116. Change "becomes" to 'became'.

L117. When referring to Fig. 4 here and below, specify the panels (a to e).

L146. Change "net production" to 'net CH₄ production'.

L166. Delete "of".

L170. Magnitude

L190. Delete "nearly".

L199. What is dashed line in Fig. 6 at 10m? Add to legend.

L219. I am very glad the authors have added Table S2, and have added an error term to these fluxes. It would be worth referring to Table S2 here, to remind readers that the error is mostly linked to air water gas exchange estimates, but that these estimates are conservative. I suggest adding a sentence such as "Our analysis of uncertainties detailed in table S2 indicates that error associated with our estimate of Pnet-s (110 ± 60 nmol L⁻¹ d⁻¹) is largely due to estimates of air-water flux, but that as described above, these estimates are conservative".

L223. You didn't mention MOX here. Worth quickly mentioning and referring to table 2.

L263. Is this 20 per mille in the lake or bottle?

L280-281. Combine these paragraphs into one.

L286. Sentence unclear. Are you trying to say that 'relying on correlations of CH₄ concentration with other variables alone is misleading, and that the production rates used here provide a clearer picture of the vertical zones important in sustaining CH₄ emissions'?

L310. Reword last sentence to '...suggest a local "compensation" with production of isotopically-depleted CH₄'.

L354. Remove second "a".

L367-372. Worth reiterating here that lateral transport in SML are much lower.

L672. Change title to 'Uncertainty assessments'.

Reviewer #2:

Remarks to the Author:

Reviewer 2. While I remain somewhat skeptical that 80% of the methane emissions from this lake are produced in the surface waters, I think that the authors have done a good job with their rebuttal and the revision of the manuscript and I support its publication without further revision. This is certainly a hot topic, and time will tell if this provocative study weathers the scrutiny I am sure it will receive.

Reviewer #3:

Remarks to the Author:

The revised manuscript has improved substantially from previous version and my concerns have been satisfactorily addressed. I consider the revised manuscript to be publishable in Nature Communications.

Sincerely

Frank Keppler

REVIEWERS' COMMENTS:

Reviewer #1 (Remarks to the Author):

The authors have put in a great deal of effort to address reviewer concerns, and I think the manuscript is in excellent shape. I only have a few minor comments:

We thank this, and all, reviewers for the excellent feedback in improving our manuscript!

L64. The sentence is a bit awkward, I would rephrase to something like "... is in direct contact with the atmosphere, implying that lake surface waters may be an important, but overlooked site of CH₄ production".

We now changed as suggested in L65 -67. "This significant source of CH₄ is in direct contact with the atmosphere, implying that lake surface waters may be an important but overlooked CH₄ production site."

L116. Change "becomes" to 'became'.

Adjusted, now in L123.

L117. When referring to Fig. 4 here and below, specify the panels (a to e).

We now add the specific panels in Fig 4 throughout L125-129, and throughout.

L146. Change "net production" to 'net CH₄ production'.

This is done, now L153.

L166. Delete "of". Agreed , L171.

L170. Magnitude. The s was removed form "magnitudes", now L175.

L190. Delete "nearly". Done , L195.

L199. What is dashed line in Fig. 6 at 10m? Add to legend. We now add to Figure 6 legend that : "The dashed line indicates the lower limit of zone 1."

L219. I am very glad the authors have added Table S2, and have added an error term to these fluxes. It would be worth referring to Table S2 here, to remind readers that the error is mostly linked to air water gas exchange estimates, but that these estimates are conservative. I suggest adding a sentence such as "Our analysis of uncertainties detailed in table S2 indicates that error associated with our estimate of

P_{net-s} ($110 \pm 60 \text{ nmol L}^{-1} \text{ d}^{-1}$) is largely due to estimates of air-water flux, but that as described above, these estimates are conservative”.

As suggested we add to L226-228: “While our analysis of uncertainties detailed in Supplementary Table 2 indicates that the error associated with $P_{\text{net-s}}$ ($110 \pm 60 \text{ nmol L}^{-1} \text{ d}^{-1}$) is largely due to air-water exchange estimates that, as described above, are conservative.”

L223. You didn’t mention MOX here. Worth quickly mentioning and referring to table 2.

We now add on L225-226 : “The mass balance includes MOx rates from in situ incubations (Table 2), although these are negligible compared to the surface losses of CH₄ to the atmosphere.”

L263. Is this 20 per mille in the lake or bottle? As now reported in L258 “ *in situ* bottle incubations”

L280-281. Combine these paragraphs into one. Now they are (L281)

L286. Sentence unclear. Are you trying to say that ‘relying on correlations of CH₄ concentration with other variables alone is misleading, and that the production rates used here provide a clearer picture of the vertical zones important in sustaining CH₄ emissions’?

This is correct-we rephrase as: L281 - 284 ” Therefore, relying on correlations of CH₄ concentration with other variables alone is misleading, while using the production rates for correlations provides a clearer picture of each vertical zone’s importance in sustaining CH₄ emissions.”

L310. Reword last sentence to ‘...suggest a local “compensation” with production of isotopically-depleted CH₄’. In L308 we add “isotopically lighter CH₄”

L354. Remove second “a”. This was done, L356

L367-372. Worth reiterating here that lateral transport in SML are much lower.

We add to L366 “although less important in the SML than at increasing depths.”

L672. Change title to ‘Uncertainty assessments’. This was done, now L668

Reviewer #2 (Remarks to the Author):

Reviewer 2. While I remain somewhat skeptical that 80% of the methane emissions from this lake are produced in the surface waters, I think that the authors have done a good job with their rebuttal and the revision of the manuscript and I support its publication without further revision. This is certainly a hot topic, and time will tell if this provocative study weathers the scrutiny I am sure it will receive.

Reviewer #3 (Remarks to the Author):

The revised manuscript has improved substantially from previous version and my concerns have been satisfactorily addressed. I consider the revised manuscript to be publishable in Nature Communications.

Sincerely